SPECIAL ISSUE
CELL BIOLOGY OF THE NUCLEUS

# Selective disruption of microtubule formation at the nuclear envelope impairs the bone resorption capacity of osteoclasts

Silvia Vergarajauregui[1,‡], Samantha Panea[1,*], Jakob O. Oltmanns[1], Ulrike Steffen[2] and Felix B. Engel[1,‡]

## ABSTRACT

Microtubule organization plays a central role in cell differentiation, orchestrating essential processes such as cell polarization, mechanotransduction, organelle positioning and intracellular transport. A hallmark of many differentiated cells is the transition from a centrosomal to a non-centrosomal microtubule-organizing center (MTOC). Here, we demonstrate that both centrosomal and nuclear envelope (NE)-associated MTOCs coexist in osteoclasts. We show that the key players for NE-MTOC formation, the AKAP6 and nesprin-1 (SYNE1) isoforms AKAP6β and nesprin-1α, previously considered muscle specific, are upregulated during osteoclast differentiation, suggesting a conserved role in NE-MTOC assembly across cell types. Targeted depletion of AKAP6 in RAW264.7-derived osteoclasts led to the displacement of the Golgi and MTOC-associated proteins PCM1, pericentrin and CDK5RAP2 from the NE, while their centrosomal localization remained intact. This selectively impaired microtubule nucleation from the NE without disrupting centrosomal microtubule activity, enabling a functional dissection of the two MTOCs. Loss of NE-MTOC activity, through AKAP6 depletion, impaired podosome formation and significantly reduced bone resorption capacity, highlighting the distinct and essential role of NE-derived microtubules in osteoclast function.

KEY WORDS: AKAP6β, Nesprin-1α, Microtubule-organizing center, Podosome, Bone resorption, Osteoclast

## INTRODUCTION

Microtubules (MTs), as part of the cytoskeleton, are integral to cellular stability, movement and polarization, as well as vesicle trafficking. MT nucleation and outgrowth occurs at MT organizing centers (MTOCs) which differ dependent on the cell type, species and cellular state. In proliferative cells, the centrosome acts as the main MTOC. Centrosomes consist of two centrioles surrounded by the pericentriolar matrix (PCM). This PCM consists of proteins such as pericentrin (PCNT), AKAP9 (also known as AKAP450) and CDK5RAP2, which are essential for MT nucleation (Bornens, 2002, 2012). During cell differentiation, in order to enable specialized cell functions, cells undergo drastic reorganization of their MT network utilizing non-centrosomal MTOCs (Becker et al., 2020). Notably, cells usually nucleate MTs from multiple MTOCs such as the centrosome, Golgi, nuclear envelope (NE) and/ or cell membrane (Sanchez and Feldman, 2017). Although it has been previously shown that MTs are important for a variety of cellular functions (Logan and Menko, 2019), it remains unclear which MTOC is responsible for which cellular function, as inhibitors of MT polymerization have been utilized, such as nocodazole or taxol, which affect MTs independently of their origin or MTOC.

Despite their importance, non-centrosomal MTOCs are poorly studied and understood. Recently, we have characterized in detail the MTOC at the NE, which is present in cardiomyocytes, skeletal muscle cells, and osteoclasts (Becker et al., 2020, 2021; Vergarajauregui et al., 2020; Zebrowski et al., 2015). The generation of a NE-MTOC requires the AKAP6 isoform AKAP6β (hereafter just AKAP6) for the recruitment of MTOC proteins to the NE (Becker et al., 2021; Vergarajauregui et al., 2020). In muscle cells, AKAP6, traditionally considered a signaling hub for protein kinase A and other signaling proteins (Passariello et al., 2015), localizes to the NE via binding to nesprin-1α, a muscle-specific isoform of nesprin-1 (SYNE1) (Pare et al., 2005) and component of the linker of nucleoskeleton and cytoskeleton (LINC) complex (Zhou et al., 2018). AKAP6 binds γ-TuRC-binding proteins containing the pericentrin and AKAP450 centrosomal targeting (PACT) domain, AKAP9 and PCNT, anchoring them to the NE and supporting MT nucleation and growth. Depletion of AKAP6 disrupts the NE-MTOC, releasing centrosomal proteins and detaching MTs from the NE (Vergarajauregui et al., 2020). Furthermore, AKAP6 and AKAP9 act as a platform tethering the Golgi to the NE, with perturbation of this interaction leading to Golgi displacement and fragmentation (Vergarajauregui et al., 2020).

Bone is a dynamic and adaptive tissue that undergoes continuous remodeling to maintain its structural integrity and respond to mechanical demands (Ansari and Sims, 2020). This process relies on a delicate balance between bone formation by osteoblasts and bone resorption by osteoclasts. Osteoclasts are large multinucleated cells derived from macrophage-like precursors (Florencio-Silva et al., 2015). The cytoskeleton of osteoclasts, composed of actin filaments, intermediate filaments and MTs (Pollard and Goldman, 2018), plays an essential role in osteoclast movement, fusion, and ultimately, bone resorption (Blangy et al., 2020). Osteoclasts accomplish bone resorption by clustering F-actin filaments into podosomes (Blangy et al., 2020; Itzstein et al., 2011), allowing them

[1]Experimental Renal and Cardiovascular Research, Department of Nephropathology, Institute of Pathology and Department of Cardiology, Friedrich-Alexander-Universität Erlangen-Nürnberg (FAU), Kussmaulallee 12, 91054 Erlangen, Germany. [2]Department of Internal Medicine 3, Rheumatology and Immunology, Friedrich-Alexander-University Erlangen-Nürnberg (FAU) and Universitätsklinikum Erlangen, 91054 Erlangen, Germany.
*Present address: Institute for Research in Biomedicine (IRB Barcelona), The Barcelona Institute of Science and Technology, 08028 Barcelona, Spain.

‡Authors for correspondence (felix.engel@uk-erlangen.de; silvia.vergarajauregui@uk-erlangen.de)

S.V., 0000-0002-9247-6123; U.S., 0000-0003-2934-3202; F.B.E., 0000-0003-2605-3429

to tightly adhere to the bone surface and form a sealing zone that creates a secluded chemical compartment called the resorption lacuna. Within this compartment, the basal cell membrane creates folds within itself creating the so-called ruffled border, where Cl⁻ channels and ATPases pump ions into the resorption lacuna, lowering the pH and thereby dissolving the mineral compounds in the bone matrix (Ng et al., 2013). In addition, lysosomal vesicles are transported towards the bone surface and release bone-degrading enzymes like cathepsin K, tartrate-resistant acid phosphatase (TRAP; also known as ACP5) and matrix metalloproteinase-9 (MMP9). The remains of the dissolved bone are resorbed and released apically (Blangy et al., 2020). MTs, which are crucial for osteoclastic functions, regulate the formation and stability of podosomes and sealing zones through interaction with F-actin filaments, helping to maintain the tight adherence of osteoclasts to the bone surface (Destaing et al., 2005). Additionally, MTs are essential for lysosomal trafficking toward the resorption lacuna, which is necessary for forming the ruffled border and enabling osteoclastic resorption (Ye et al., 2011). The strong dependence of osteoclastic functions on MTs makes their structure and organization crucially important in osteoclasts. Notably, depolymerization of F-actin as well as MTs inhibits the pit-forming and bone resorption activity of osteoclasts in a dose-dependent and reversible manner (Okumura et al., 2006), highlighting the cytoskeleton as a promising therapeutic target to modulate osteoclast activity.

A 2003 study reported that osteoclasts recruit PCNT and Golgi to the NE and nucleate MTs there (Mulari et al., 2003). Yet, in a comparative analysis of osteoclasts across species they observed intriguing differences in MT organization. In non-resorbing avian osteoclasts, MTs radiated from multiple centrosomal MTOCs. Upon activation for bone resorption, these centrosomal MTOCs disappeared, and Golgi membranes redistributed to encircle the nuclei. In contrast, mammalian osteoclasts typically lacked or possessed very few centrosomal MTOCs, with MTs predominantly nucleating from perinuclear areas enriched in PCM proteins, such as PCNT (Mulari et al., 2003). Recent studies confirmed that mammalian osteoclasts establish MT nucleation sites at nuclear surfaces (Vergarajauregui et al., 2020; Yamamoto et al., 2019), similar to multinucleated myotubes and cardiomyocytes.

In contrast to the observation of Mulari et al. that MTs radiate from centrosomal MTOCs only in non-resorbing avian osteoclasts, Philip et al. reported that osteoclasts organize centrosomes into large clusters for MT nucleation and bone resorption independently of their polarization (Philip et al., 2022). In order to determine whether centrosomal or centrosome-like MTOCs and/or NE-MTOCs are required for bone resorption, we depleted AKAP6 in RAW264.7-derived osteoclasts and assessed the effect on NE-MTOC and centrosomal or centrosome-like MTOC function, podosome or F-actin ring formation and bone resorption. Here, we show that depletion of AKAP6 in RAW264.7-derived osteoclasts disrupted the NE-MTOC but not the centrosomal or centrosome-like MTOC, causing an impairment of podosome formation and ultimately a reduction of bone resorption capacity.

## RESULTS
### MTOC proteins and Golgi are recruited to the nuclear envelope in RAW264.7-derived osteoclasts
Previously, it has been reported that osteoclasts contain centrosomal MTOCs as well as NE-MTOCs and there are claims for both MTOC types to be responsible for the podosome formation and/or

resorption activity of osteoclasts (Mulari et al., 2003; Philip et al., 2022; Vergarajauregui et al., 2020). Philip et al. based their conclusion that "osteoclasts organize centrosomes into large clusters for microtubule nucleation and bone resorption" mainly on experiments with RAW264.7 cells, which were treated with centrosome declustering agents (dynarrestin, a blocker of cytoplasmic dynein, CW069, a small-molecule inhibitor of KIFC1, and griseofulvin, an antimitotic agent that reduces MT dynamicity) (Philip et al., 2022). Notably, blocking dynein function has been shown to affect NE-MTOC-mediated nuclei alignment in skeletal muscle cells (Espigat-Georger et al., 2016). Furthermore, sliding of centrosome-unattached MTs finetunes the trajectory of neuronal migration utilizing KIFC1 as molecular motor regulating this sliding (Muralidharan et al., 2022). Consequently, it cannot be excluded that dynarrestin, CW069 and griseofulvin also affect NE-MTOC-MTs, or MT function in osteoclasts.

To address this controversy, the MTOCs from RAW264.7-derived osteoclasts after 4 days of RANKL addition were characterized. To confirm the presence of a NE-MTOC, we examined the localization of AKAP6, a key protein involved in the formation of the NE-MTOC (Becker et al., 2021; Vergarajauregui et al., 2020). As shown in Fig. 1A, we confirmed the expression of AKAP6 in RAW264.7-derived osteoclasts exclusively at the NE. Notably, expression of AKAP6 and nesprin-1α are known to be sufficient to recruit MTOC proteins and the Golgi to the NE (Vergarajauregui et al., 2020). Consistent with this, GM130 (also known as GOLGA2), a marker for the Golgi, colocalized around the NE (Fig. 1A). In addition, we confirmed the presence of PCNT and CDK5RAP2 at the NE (Fig. 1B,C; Fig. S1A), which are MTOC proteins crucial for MT nucleation (Lin et al., 2015), as previously observed in human osteoclasts, skeletal muscle cells, and cardiomyocytes (Becker et al., 2021; Mulari et al., 2003; Vergarajauregui et al., 2020). PCNT, as recently described (Philip et al., 2022), as well as CDK5RAP2 also localized to individual and clustered centrosomes (Fig. 1B,C; Fig. S1). These data indicate that RAW264.7-derived osteoclasts contain both centrosomal MTOCs and NE-MTOCs.

Finally, we stained for PCM1 as NE-MTOCs in skeletal muscle cells and cardiomyocytes are characterized by the fact that PCM1 can only be detected at the NE but not at the centrosome (Becker et al., 2021; Vergarajauregui et al., 2020). Staining of RAW264.7-derived osteoclasts showed no colocalization of PCM1 with the centrosomal marker γ-tubulin. In contrast, PCM1 was readily detected at the NE (Fig. 1D). These data suggest that MTs nucleated at the NE-MTOC might be as important in osteoclasts as in cardiomyocytes, in which NE-MTOC MTs control hypertrophy, or in skeletal muscle cells, in which NE-MTOC MTs control nuclear positioning. Notably, although AKAP6 and PCM1 are present at the nuclear envelope of nearly all nuclei, the recruitment of PCNT and CDK5RAP2 is more heterogeneous. Only 82.33±13.5% or 82.67±4.0% (mean±s.d., $n$=3) of nuclei show enrichment of CDK5RAP2 or PCNT, respectively. We recently demonstrated that the recruitment of centrosomal proteins to the nuclear envelope in differentiating myoblasts occurs in a gradual and sequential manner (Das et al., 2025). Similarly, the differential recruitment of centrosomal proteins to the NE in RAW264.7-derived osteoclasts might reflect variable stages of nuclear maturation and/or progressive establishment of NE-MTOCs within individual nuclei.

Collectively, these findings suggest that the organization of the NE-MTOC in RAW264.7-derived osteoclasts follows the same pattern as that in skeletal muscle cells and cardiomyocytes based on the presence of MTOC proteins.

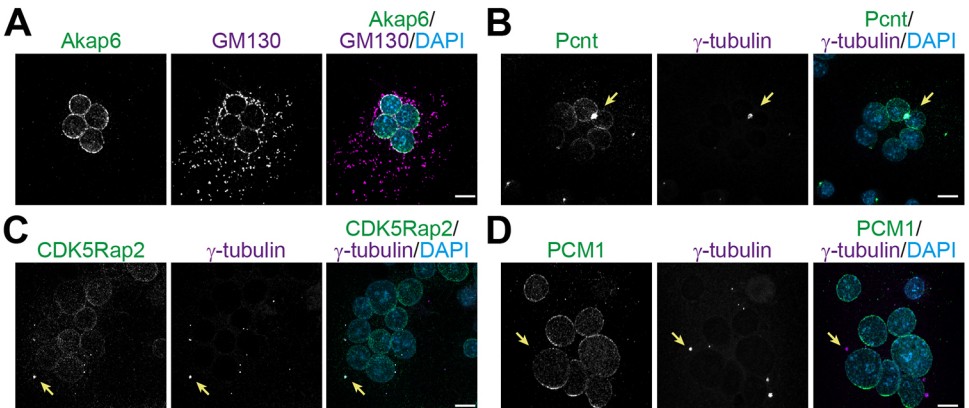

**Fig. 1. MTOC proteins and Golgi are recruited to the nuclear envelope in RAW264.7-derived osteoclasts.**
(A) Immunostaining of AKAP6 (green) and GM130 (magenta) together with DNA (DAPI) in RAW264.7-derived osteoclasts after 4 days of RANKL addition. Scale bars: 10 µm. Immunostaining of (B) PCNT (green), (C) CDK5RAP2 (green) and (D) PCM1 (green) and γ-tubulin (magenta) together with DNA (DAPI) in RAW264.7-derived osteoclasts after 4 days of RANKL addition. Arrows indicate centrosomes. Scale bars: 10 µm. Images representative of three experimental repeats.

## RAW264.7-derived osteoclasts contain multiple MTOC types

To determine whether NE-MTOCs in RAW264.7-derived osteoclasts are actively nucleating MTs, we assessed the steady-state organization of MTs by staining α-tubulin. MTs in RAW264.7-derived osteoclasts accumulated around the nucleus, which was positive for AKAP6, as well as GM130, a marker for Golgi (Fig. 2A, asterisks). In addition, MTs in aster-like structures were observed (Fig. 2A, arrows) indicating the existence of centrosomal or centrosome-like MTOCs.

MTs can be transported and anchored at other locations (Bartolini and Gundersen, 2006). Thus, to assess where the MTs are indeed nucleated, we performed MT outgrowth assays. RAW264.7-derived osteoclasts were treated with nocodazole for 1 h to depolymerize MTs (Fig. 2B, upper panel), followed by nocodazole washout, and then allowed to recover for 1 to 2 min at 37°C. Cells were then stained for EB1 (also known as MAPRE1), a marker for the plus-end of growing MTs. EB1 localized prominently at the NE in 94±2.4% (mean±s.d., n=3) of the osteoclast nuclei, indicating that MTs were indeed growing from this site (Fig. 2B, asterisks). In addition to EB1 localization at the NE, EB1 was found in aster-like formations, colocalizing with the centriole marker γ-tubulin (Fig. 2B, arrow). These data further support the existence of both

centrosomal or centrosome-like (single or more than two centrioles) MTOCs and NE-MTOCs.

Previously, it has been shown that the Golgi can also act as an MTOC (Sanders and Kaverina, 2015). As the Golgi localizes in RAW264.7-derived osteoclasts to the NE, similar to what is seen in skeletal muscle cells and cardiomyocytes, we wondered whether nuclei exist in which MTs can be observed at the NE in the absence of the Golgi. For this purpose, we examined the distribution of GM130, as a marker of Golgi, and PCNT, which nucleates MTs from the NE in cardiomyocytes (Vergarajauregui et al., 2020), along with α-tubulin in nocodazole-washout osteoclasts. As seen in Fig. 2C, although most of the nuclei were positive for both PCNT and GM130 (85±0.09%; mean±s.d., n=3), we observed some nuclei (12±0.04%; n=3) that exhibited MT outgrowths that were positive for PCNT but lacked GM130. This indicates that the NE of RAW264.7-derived osteoclasts can recruit MTOC proteins crucial for MT nucleation in the absence of the Golgi, as previously also shown for cardiomyocytes (Vergarajauregui et al., 2020). These results confirm that osteoclasts possess an active NE-MTOC that is capable of nucleating MTs, in addition to centrosomal or centrosome-like MTOCs.

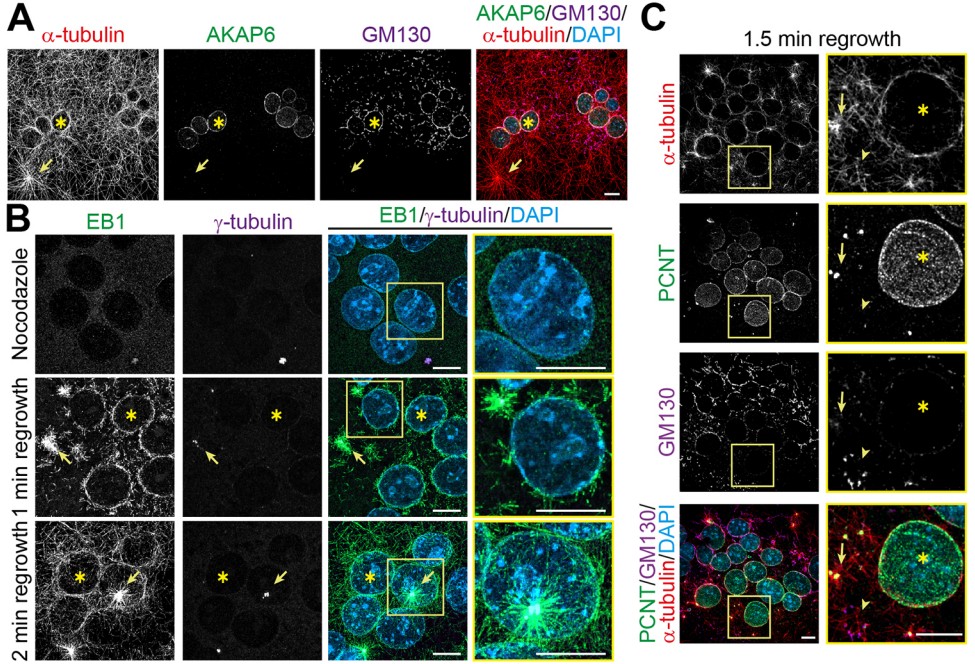

**Fig. 2. RAW264.7-derived osteoclasts exhibit a nuclear envelope MTOC alongside a functional centrosome.**
(A) Immunostaining of α-tubulin (red), AKAP6 (green), GM130 (magenta) and DNA (DAPI) in RAW264.7-derived osteoclasts differentiated for 4 days. (B) Immunostaining of EB-1 (green), γ-tubulin (magenta) and DNA (DAPI) in RAW264.7-derived osteoclasts treated with nocodazole (upper row) and after 1 min (middle row) or 2 min (bottom row) recovery at 37°C. In A and B, asterisks indicate the perinuclear microtubule cage and arrows denote centrosomal asters. (C) Immunostaining of α-tubulin (red), PCNT (green), GM130 (magenta), and DNA (DAPI) in 4 day RAW264.7-derived osteoclasts after 1.5 min of recovery from nocodazole-induced microtubule depolymerization. Asterisks denote the NE nucleation, arrows denote centrosomal nucleation and arrowheads denote Golgi nucleation. Scale bars: 10 µm. Images representative of three experimental repeats.

To evaluate the sensitivity of centrosomal or centrosome-like MTOCs and NE-MTOCs to pharmacological drugs that perturb the clustering of centrosomes, we treated osteoclasts with the agent griseofulvin (GF). GF has been reported to inhibit centrosome clustering in osteoclasts resulting in impaired F-acing ring formation and bone resorption (Philip et al., 2022). However, although GF has been reported to interfere with centrosome clustering (Philip et al., 2022; Rebacz et al., 2007), it is known that it affects the dynamic instability of MTs (Panda et al., 2005). Consistent with this, GF treatment of RAW264.7-derived osteoclasts led to a noticeable redistribution of MTs, resulting in shorter, less organized filaments attached to both centrosomal and NE regions (Fig. S2). Furthermore, PCNT staining at the NE shifted from a relatively homogeneous distribution to a more punctate, discontinuous pattern, indicating altered NE-MTOC architecture. These results suggest that pharmacological agents used to disrupt centrosome clustering might affect both centrosomal and nuclear envelope-associated MTOCs. Therefore, alternative approaches will be necessary to distinguish the specific functional contributions of the two different MTOCs.

### Nesprin-1α and AKAP6 build the NE-MTOC anchor in RAW264.7-derived osteoclasts

The NE-MTOC anchor is composed of nesprin-1α and AKAP6 (Becker et al., 2021; Gimpel et al., 2017; Vergarajauregui et al.,

2020). The nesprin-1α is a smaller nesprin-1 isoform expressed almost exclusively in skeletal muscle and heart (Duong et al., 2014). Nesprin-1α has a unique short sequence at its N-terminus, but is otherwise identical to the C-terminal region of nesprin-1-giant (Holt et al., 2016). Nesprin-1α is required for AKAP6 localization at the NE (Pare et al., 2005), and together these proteins are sufficient to induce recruitment of MTOC proteins to the NE of non-muscle cells (Vergarajauregui et al., 2020). Moreover, myogenin expression selectively induces nesprin-1α and AKAP6 leading to NE-MTOC formation in fibroblasts (Becker et al., 2021).

To characterize the NE-MTOC anchor in RAW264.7-derived osteoclasts, we examined the expression and localization of nesprin-1α together with AKAP6 by immunostaining in undifferentiated and osteoclast-differentiated cells. As the isoform-specific antibody for nesprin-1α does not recognize the mouse variant, we used an antibody detecting both nesprin-1 and nesprin-1α. Immunostaining analysis revealed that undifferentiated RAW264.7 cells (upper panels) lacked detectable nesprin-1 and AKAP6 at the nuclear envelope (Fig. 3A, upper panels). In contrast, upon RANKL-induced differentiation, nesprin-1 and AKAP6 became robustly enriched at the NE of multinucleated osteoclasts (Fig. 3A, lower panels), whereas smaller mononucleated cells characteristic for undifferentiated RAW264.7 remained negative (Fig. 3A, arrows). To further assess their expression during osteoclastogenesis, we

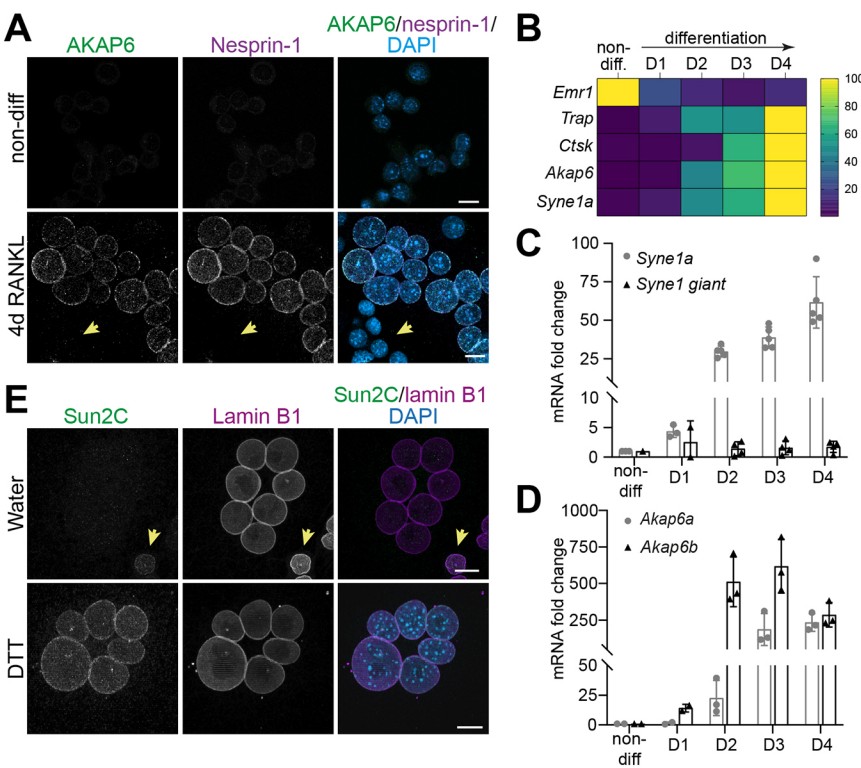

**Fig. 3. Nesprin-1α and AKAP6β isoforms predominate in RAW264.7-derived osteoclasts and form a functional LINC complex.** (A) Immunostaining of AKAP6 (green), nesprin-1 (magenta) and DNA (DAPI) in RAW264.7 cells either non-differentiated (upper panels) or differentiated into osteoclasts for 4 days with 50 ng ml⁻¹ of RANKL (lower panels). Non-differentiated cells lack detectable expression of both AKAP6 and nesprin-1 (also denoted with an arrow). Scale bars: 10 µm. (B) Heatmap of qPCR-derived RNA expression levels for *Emr1*, *Akap6*, *Syne1a*, *Trap* and *Ctsk* during osteoclast differentiation (days 1–4 of RANKL treatment) relative to non-differentiated precursor cells (*n*=3). Expression values are normalized and represented as a percentage of maximal expression per gene using the viridis color map. (C) qPCR analysis of *Syne1a* (gray circles) and *Syne1 giant* (black triangles) mRNA expression during osteoclast differentiation (D1, D2, D3, D4, day 1–4). (D) qPCR analysis of *Akap6a* (gray circles) and *Akap6b* (black triangles) transcript levels at the indicated differentiation stages. Data in C and D are presented as mean±s.d. from three independent experiments, normalized to the non-differentiated condition. (E) Representative images of RAW264.7-derived osteoclasts treated with vehicle (water) or 5 µM DTT and co-stained with Sun2C (green), laminB1 (magenta) antibodies and DAPI. The Sun2C antibody recognizes an epitope which is masked when nesprin-1α is upregulated during osteoclast differentiation. DTT treatment restores epitope accessibility. Non-differentiated cells, indicated with an arrow, exhibit Sun2C reactivity. Scale bars: 10 µm. Images in A and E representative of three experimental repeats.

quantified transcript levels over 4 days of RANKL treatment. As expected, mRNA levels of the macrophage marker *Emr1* (also known as *Adgre1*) decreased, whereas mRNA levels of osteoclast markers *Trap* and *Ctsk* were strongly upregulated (Fig. 3B). Notably, both *Akap6* and *Syne1a* displayed a similar differentiation-dependent increase, with *Akap6* increasing more than 200-fold and nesprin-1α ~60-fold relative to the levels in undifferentiated cells. To assess nesprin-1 isoform-specific transcript usage during osteoclastogenesis, we performed quantitative (q)PCR using isoform-specific primers targeting *Syne1 giant* and *Syne1a*. We found that *Syne1a* was strongly upregulated starting on day 1, whereas the levels of the *Syne1 giant* isoform remained largely unchanged (Fig. 3C). Similar to nesprin-1, AKAP6 exists in two isoforms, the larger AKAP6α form, which is predominantly expressed in the brain, and the shorter AKAP6β form, which is expressed in skeletal muscle and heart (Michel et al., 2005). Isoform-specific transcript analysis showed that both Akap6 isoforms were induced during differentiation, but the short muscle-associated *Akap6b* increased earlier and more robustly than *Akap6a* (Fig. 3D). These data indicate that the nesprin-1α–AKAP6β module, previously considered to be muscle restricted, is specifically induced during osteoclast differentiation.

Nesprin-1 binds to SUN proteins in the NE to form the LINC complex. Previous work by Sharma et al. has demonstrated that during myoblast differentiation into myotubes, SUN2 undergoes a conformational change upon nesprin binding (Sharma and Hetzer, 2023). This conformational change masks a specific epitope in SUN2, preventing its detection by the antibody SUN2C, which can recognize this epitope only in its unmasked state in myoblasts. Treatment with DTT, which reverses the conformational change, restores SUN2C antibody recognition. We applied this approach to assess whether a similar conformational change occurs during osteoclast differentiation. In differentiated RAW264.7-derived osteoclasts, the SUN2C antibody failed to recognize SUN2 (Fig. 3E), suggesting that a similar conformational change occurs in osteoclasts to that reported for skeletal muscle cells. Supporting this, SUN2 staining by the SUN2C antibody was restored after DTT treatment, and SUN2 was detected in the smaller nuclei, characteristic for undifferentiated cells, without DTT treatment (Fig. 3E, arrow).

Taken together, these results demonstrate that *Syne1a* and *Akap6b*, previously thought to be restricted to muscle tissues, are also expressed in osteoclasts. This suggests that these isoforms are not muscle specific but rather are also expressed in cell types that assemble an NE-MTOC. Furthermore, the observation of a SUN2 conformational change during osteoclast differentiation supports the existence of a conserved mechanism for NE-MTOC anchoring that relies on the nesprin-1α–AKAP6β complex, likely through the LINC complex, across different cell types.

## AKAP6 is required as anchor for NE-MTOCs and Golgi in RAW264.7-derived osteoclasts

To validate that AKAP6 is required to anchor NE-MTOCs and organize the Golgi in RAW264.7-derived osteoclasts, as previously shown in primary osteoclasts (Vergarajauregui et al., 2020), we performed AKAP6 knockdown using siRNA and evaluated the localization of nesprin-1, the Golgi and MTOC proteins. At 3 days after transfection of siRNA targeting *Akap6* (siAKAP6), performed after 1 day of RANKL treatment, qPCR showed that *Akap6* transcript levels were reduced by 74.33±2.52% (mean±s.d.), while *Syne1a* expression remained unchanged (Fig. 4A). Furthermore, osteoclast differentiation markers, including *Trap*, *Ctsk* and *Nfatc1*,

and the macrophage marker *Emr-1* were unaffected by *Akap6* knockdown, indicating that osteoclast differentiation was not impacted (Fig. 4A). Immunostaining analysis revealed a marked reduction of AKAP6 protein in 98.33±2.89% (mean±s.d.; *n*=3) of osteoclast nuclei compared to siControl-transfected cells (Fig. 4C). In contrast, nesprin-1 localization remained unaffected in 92.0±1.0% of nuclei (Fig. 4D), supporting its role as the upstream anchor for AKAP6 at the NE. Analysis of the Golgi, which normally accumulates around the nucleus, showed that nuclear-associated Golgi structures were absent in 94.33±4.04% of nuclei, and instead Golgi structures were dispersed throughout the cytoplasm, indicating that its NE localization is AKAP6 dependent (Fig. 4C,E). PCM1 was absent from the NE in 98.33±1.16% in AKAP6-depleted nuclei and did not relocalize to the centrosomes (Fig. 4E). PCNT and CDK5RAP2 were also lost from the NE in 96.00±0.16% and 99.33±1.16% of nuclei, respectively, in AKAP6-depleted cells (Fig. 4F,G), however, their centrosomal localization remained unaffected. These data indicate that AKAP6 depletion selectively disrupts NE-MTOC components without affecting the integrity of centrosomal MTOCs, and provides an opportunity to dissect the specific contributions of NE-MTOCs versus centrosomal MTOCs in osteoclast function and bone resorption.

Collectively, these findings demonstrate that AKAP6 is essential for the recruitment and stabilization of MTOC components and Golgi at the NE in osteoclasts, acting downstream of nesprin-1α.

## AKAP6 is required for MTOC activity at the NE but not the centrosome

To determine whether AKAP6 depletion affects specifically the nucleation of MTs at NE-MTOCs and not centrosomal or centrosome-like MTOCs, we assessed MT outgrowth patterns in AKAP6-depleted cells. As shown in Fig. 5A, AKAP6 knockdown resulted in a loss of MT outgrowth from the NE based on anti-α-tubulin staining. This observation aligns with the absence of PCNT and dispersion of the Golgi away from the NE, indicating disrupted NE-MTOC function. Quantification of MT outgrowth from the NE by measuring the fluorescence intensity of α-tubulin in concentric bands of 0.2 μm around the nucleus confirmed our conclusion (Fig. 5B). In AKAP6-depleted cells, the α-tubulin intensity showed a significantly reduced intensity peak (1.60±0.16 versus 2.00±0.13 in control, *P*=0.0264, mean±s.d.) at the nuclear edge. Moreover, the peak position shifted further outward from the NE (0.275±0.062 μm versus 0.083±0.033 μm in control, *P*=0.0092), indicating reduced α-tubulin growth from the nuclear envelope in comparison to control cells. Quantification of PCNT levels and Golgi localization at the NE confirmed that both PCNT and Golgi staining at the NE were significantly reduced in AKAP6-depleted cells compared to controls (Fig. 5B). However, no obvious reduction of PCNT at the centrosomes was observed (Fig. 5A), consistent with the findings shown in Fig. 4F. Additionally, MT outgrowth from the centrosome (Fig. 5A, arrow) and from Golgi (Fig. 5A, arrowhead) remained detectable in both control and AKAP6-depleted cells. To further characterize centrosomal MT nucleation in AKAP6-depleted cells, we performed co-staining of α-tubulin and EB-1 together with γ-tubulin, a marker of centrosomes (Fig. 5C). Although AKAP6-depleted osteoclasts showed a clear reduction of MT outgrowth from the NE, all γ-tubulin-positive centrosomes, whether single or clustered, exhibited robust MT outgrowth in both siControl- and siAKAP6-treated osteoclasts (Fig. 5C, insets). In contrast to the reduced α-tubulin intensity at the NE of AKAP6-depleted cells, quantification of α-tubulin intensity surrounding centrosomes

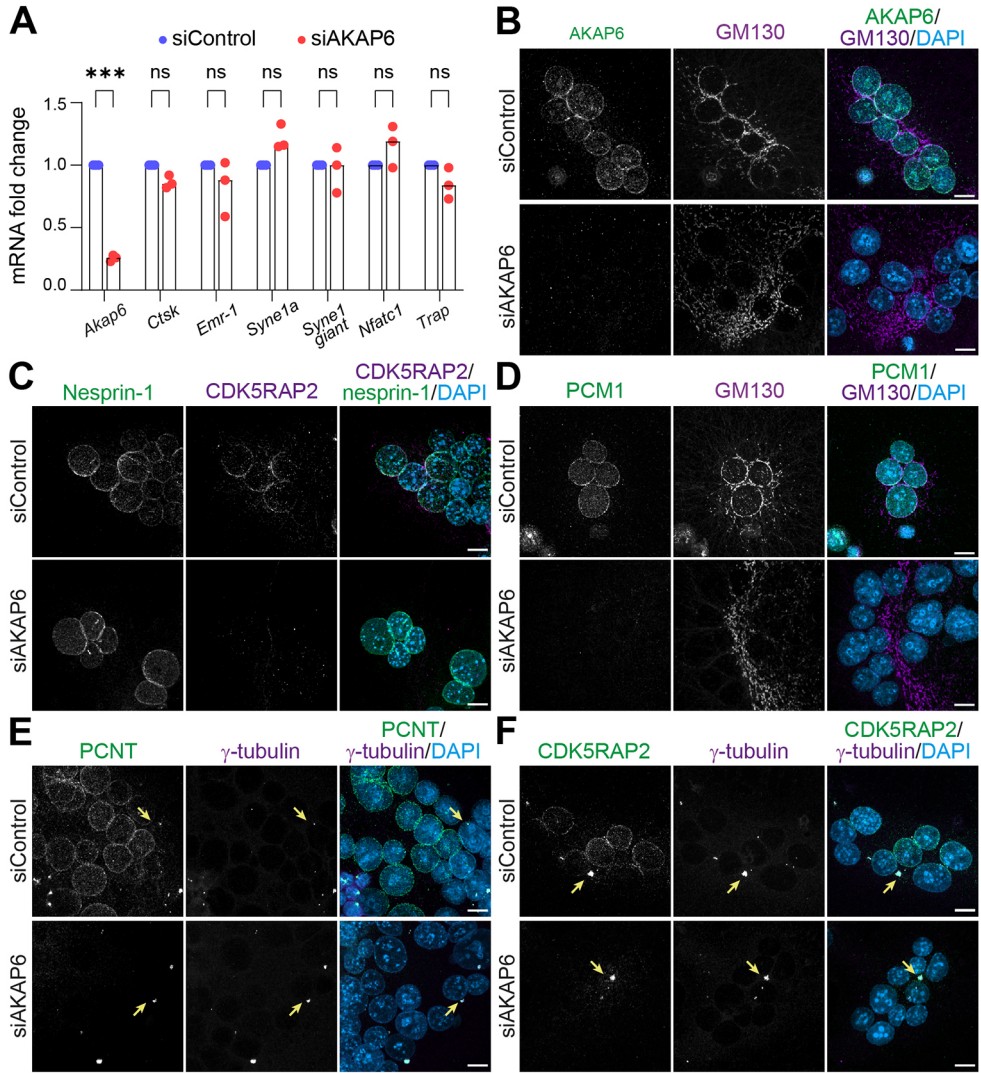

**Fig. 4. AKAP6 knockdown disrupts NE-MTOC formation in RAW-derived osteoclasts.** (A) qPCR analysis of mRNA expression in siControl- and siAKAP6-treated osteoclasts. Bars represent mean fold change (normalised to siControl) for the indicated genes, with individual experimental values shown as dots (three independent experiments). *Akap6* expression was significantly reduced following AKAP6 depletion, whereas *Syne1a*, *Syne1 giant*, the macrophage marker *Emr1*, and the osteoclast markers *Ctsk*, *Nfatc1* and *Trap* transcript levels were not significantly affected. \*\*\**P*<0.001; ns, not significant (two-way ANOVA followed by Bonferroni's multiple comparisons test). (B–F) Immunostaining in RAW264.7-derived osteoclasts transfected with control siRNA (siControl) or AKAP6 siRNA (siAKAP6) of (B) AKAP6 (green) and GM130 (magenta), and DNA (DAPI); (C) nesprin-1 (green), CDK5RAP2 (magenta), and DNA (DAPI); (D) PCM1 (green), GM130 (magenta), and DNA (DAPI); (E) PCNT, (green), γ-tubulin (magenta), and DNA (DAPI); and (F) CDK5RAP2 (green), γ-tubulin (magenta), and DNA (DAPI). Note that although perinuclear localization of PCNT and CDK5RAP2 is lost upon AKAP6 knockdown, centrosomal localization is unaffected (arrows). Scale bars: 10 µm. Images in B–F representative of three experimental repeats.

showed that AKAP6 depletion did not impair centrosomal MT nucleation (Fig. 5D). However, because centrosomes can be positioned in spatially constrained regions (e.g. between nuclei or freely in the cytoplasm), and osteoclasts exhibit heterogeneous centrosome configurations (ranging from single to clustered centrosomes), direct quantitative comparison is challenging. Taken together, the imaging and quantitative analysis demonstrate that AKAP6 depletion does not impair centrosomal MT activity.

Overall, these data demonstrate that AKAP6 depletion selectively disrupts NE-MTOC function while preserving centrosomal MTOC activity. This distinction provides a valuable tool to determine the specific roles of centrosomal or centrosome-like MTOCs and NE-MTOCs in osteoclast function and bone resorption.

## The NE-MTOC is required for podosome maturation and bone resorption

During osteoclast polarization and activation, osteoclasts form a specialized adhesion structure known as the sealing zone, which ensures firm attachment to the bone and isolates the resorption pit where proteases and protons are secreted to break down the bone matrix. The sealing zone is stabilized by a dense F-actin ring, which is regulated by MTs enriched at the periphery (Destaing et al., 2005). On non-mineralized surfaces, osteoclasts instead develop a

slightly different structure called the podosome belt. Previously, it has been shown that MT integrity is essential for podosome belt formation (Destaing et al., 2005). Given the crucial role of MTs in stabilizing the podosome belt through crosstalk with the F-actin cytoskeleton, we assessed MT distribution at the osteoclast periphery, particularly around the podosome belt margin. In control osteoclasts, MTs extended to the podosome belt and penetrated the F-actin ring, resulting in the colocalization of MTs and F-actin along the podosome belt, as previously described (Batsir et al., 2017; Biosse Duplan et al., 2014) (Fig. 6A). In contrast, in AKAP6-depleted cells, MTs failed to extend into the F-actin ring, as indicated by a marked reduction of MT–F-actin colocalization (Fig. 6A). To quantify this disruption, we employed the BIOP JACoP macro (Bolte and Cordelieres, 2006) to measure pixel colocalization between MTs and F-actin structures in the podosome belt. The belt region was manually delineated using the polygon tool in ImageJ based on the continuous F-actin ring. The overlap between the MT and F-actin channels was then normalized to the total F-actin signal within each belt region, yielding the fraction of F-actin that colocalized with MTs. AKAP6-depleted cells exhibited significantly reduced MT-F-actin colocalization at the podosome belt compared to what was seen in control RAW264.7-derived osteoclasts (Fig. 6B). These findings

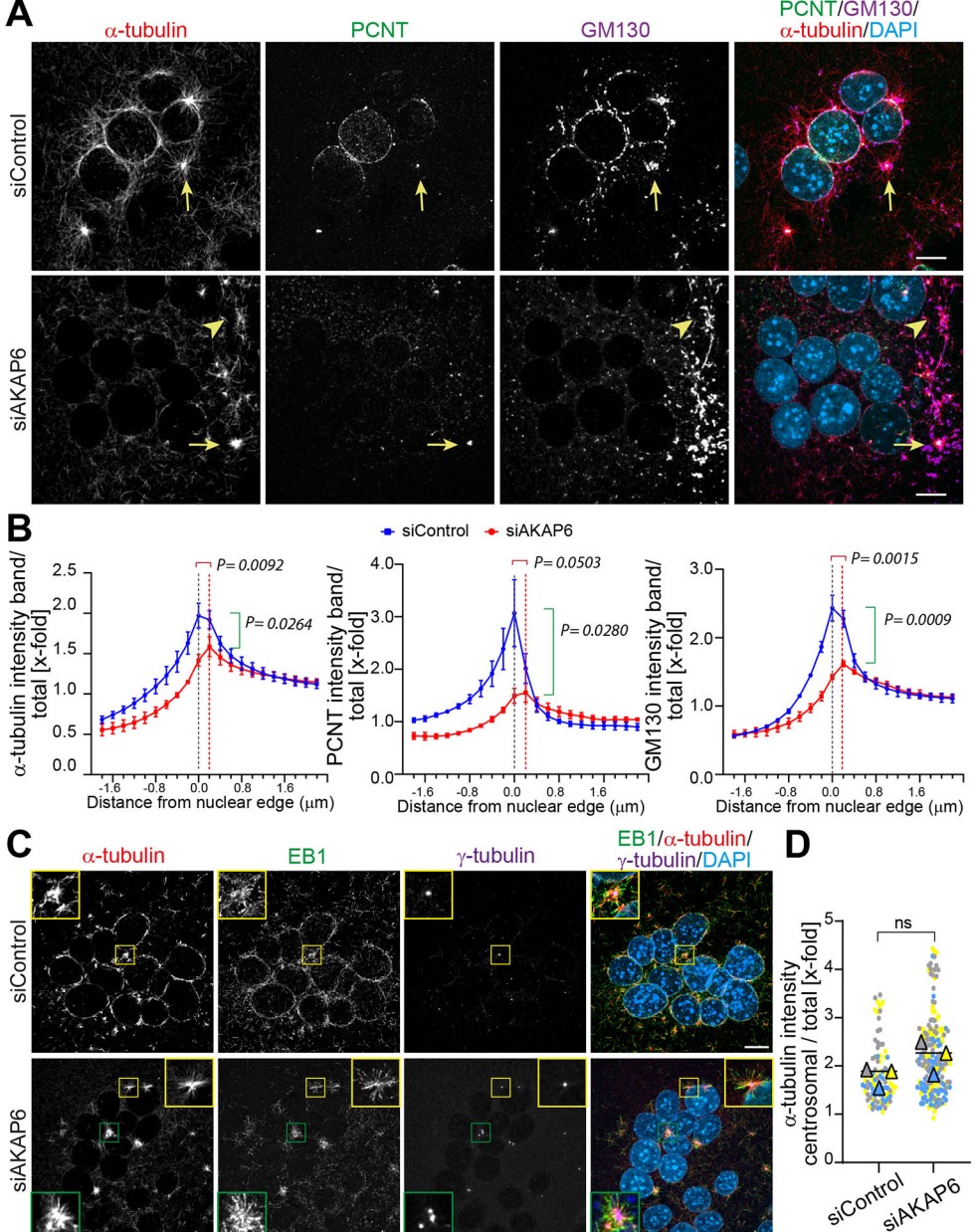

**Fig. 5. AKAP6 Depletion disrupts NE-MTOC function, but not centrosomal-MTOC.** (A) Immunostaining of α-tubulin (red), PCNT (green), GM130 (magenta) and DNA (DAPI) in RAW264.7-derived osteoclasts transfected with control siRNA (siControl) or AKAP6-targeting siRNA (siAKAP6). All stainings were performed in cells subjected to nocodazole washout (1.5 min). The yellow arrow indicates centrosomal microtubule organization and the arrowhead indicates Golgi-derived nucleation. (B) Quantification of fluorescence intensity profiles for α-tubulin, PCNT and GM130 in 0.2 μm wide concentric bands relative to distance from the nuclear edge (0.0). For each biological replicate (*n*=3), 33–36 nuclei were analyzed per condition, and graphs show the mean ±s.d. of the three independent biological experiments. The outward shift in the α-tubulin peak position is indicated by the red bracket, and the reduction in peak intensity is indicated by the green bracket. Statistical comparisons of peak position and peak intensity (amplitude) were performed at the experiment level with an unpaired two-tailed *t*-test; corresponding *P*-values are shown in the graphs. (C) Immunostaining of α-tubulin (red), EB1 (green), γ-tubulin (magenta) and DNA (DAPI) in RAW264.7-derived osteoclasts transfected with siControl or siAKAP6 performed after 1 min of nocodazole washout. Insets (2.5× magnification) highlight γ-tubulin-positive centrosomes exhibiting normal centrosomal microtubule outgrowth in both conditions. (D) Quantification of tubulin intensity in a 3 μm area surrounding each centrosome in siControl cells and siAKAP6 centrosomes from three independent experiments. SuperPlots display the overall mean (black line) and biological-replicate mean (color-coded triangles) from *n*=3, with the individual centrosomes (30–35 per condition) shown as matching color-coded dots superimposed beneath them. ns, not significant (paired two-tailed *t*-test). Scale bars: 10 μm.

suggest that AKAP6-dependent NE-MTOC organization is important for the MT–F-actin interactions required for proper podosome belt formation. To substantiate this conclusion, we examined podosome belt morphology in control and AKAP6-depleted RAW264.7-derived osteoclasts grown on glass. In control cells, phalloidin staining revealed a continuous, well-organized

F-actin ring with densely packed podosomes, F-actin-rich adhesion structures, forming a characteristic belt (Fig. 6C). In AKAP6-depleted osteoclasts, F-actin rings were still present; however, the podosomes appeared less intense, less sharply resolved, and were surrounded by a more diffuse F-actin background (Fig. 6C). A similar phenotype, characterized by a more diffuse podosome

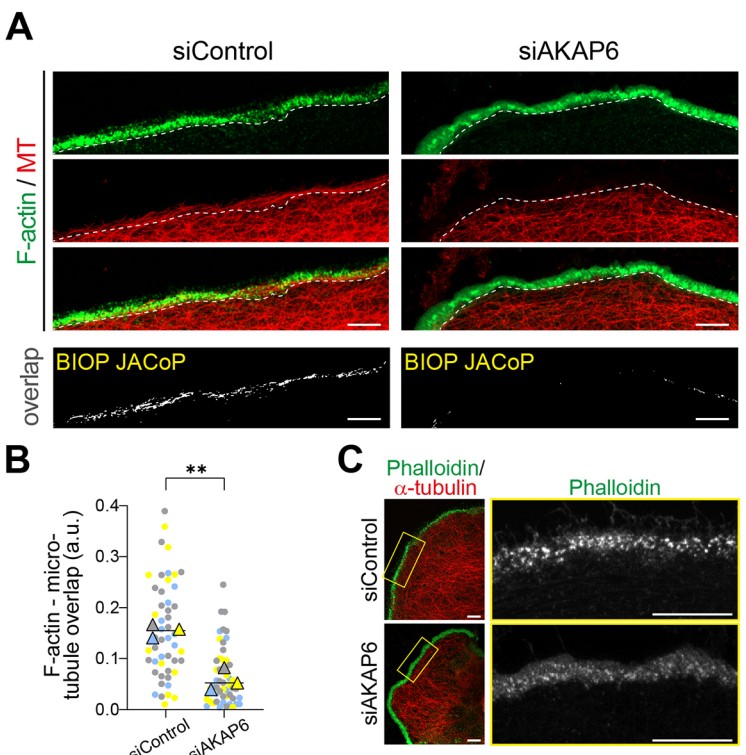

**Fig. 6. AKAP6 depletion disrupts microtubule-actin cytoskeletal crosstalk in osteoclasts.** (A) Representative images of RAW264.7-derived osteoclasts transfected with siControl or siAKAP6, stained for F-actin (phalloidin, green) and microtubules (α-tubulin, red). Corresponding BIOP JACoP analyses are shown in grayscale, illustrating the spatial overlap between F-actin and microtubules (lower panel). Scale bars: 10 µm. Dotted white lines indicate the boundary between the actin-dense sealing zone and the cytoplasm. (B) Quantification of microtubule-actin overlap in siControl and siAKAP6-treated osteoclasts. The actin-microtubule overlap was quantified in arbitrary units (a.u.) as the ratio of actin that colocalized with microtubules normalized against the total actin pixel count. SuperPlots display the overall mean (black line) and biological-replicate mean (color-coded triangles) from n=3, with the individual cells (12–25 per condition) shown as matching color-coded dots superimposed beneath them. **P=0.0043 (paired two-tailed t-test). (C) Representative immunostaining images of RAW264.7-derived osteoclasts transfected with siControl or siAKAP6 co-stained for actin (phalloidin, green) and microtubules (α-tubulin, red). The right panels show higher magnification images of the boxed regions, highlighting actin ring structure. Images in C representative of three experimental repeats. Scale bars: 10 µm.

organization, was previously reported upon Tubb6 depletion (Maurin et al., 2021), supporting the interpretation that perturbations in MT organization can impair podosome maturation.

Podosome belt abnormalities observed on glass surfaces suggest that osteoclasts might also exhibit defects in their ability to form a proper sealing zone, leading to impaired resorptive function on mineralized substrates. To directly measure the impact of AKAP6 depletion on osteoclast bone resorption activity, we cultured control and AKAP6-depleted RAW264.7-derived osteoclasts on bone-mimetic calcium phosphate (CaP) surfaces and quantified bone resorption. von Kossa staining revealed a ∼50% reduction in total resorption area in AKAP6-depleted osteoclasts compared with controls (Fig. 7A,B). However, because von Kossa staining is incompatible with podosome visualization, this approach does not distinguish whether reduced resorption results from fewer active osteoclasts or diminished resorptive capacity per cell. To overcome this limitation, we adapted a protocol from Cen et al. (2022), in which sealing zones are labeled with phalloidin, and CaP is fluorescently labeled with calcein, allowing simultaneous visualization of sealing zones and resorption pits by fluorescence microscopy.

Both control and AKAP6-depleted RAW264.7-derived osteoclasts formed F-actin-rich sealing zones when plated on CaP, as confirmed by phalloidin staining (Fig. 7C). However, the functional outcomes differed markedly. Control RAW264.7-derived osteoclasts effectively resorbed the CaP surface, as evidenced by reduced calcein staining within resorption pits (Fig. 7C). In contrast, most sealing zones in AKAP6-depleted cells failed to resorb CaP, retaining high calcein staining levels indicative of poor resorption activity (Fig. 7C). To quantify these differences, we measured sealing zone size and calculated a calcein ratio, by normalizing the calcein intensity inside the sealing zone (phalloidin-positive area) to the intensity outside the sealing zone. A ratio below 1 indicates resorption (Fig. 7D). In control osteoclasts, the resorption activity increased with sealing zone size.

Only small sealing zones ($<500$ µm$^2$) showed limited resorption, whereas $81.7\pm16.9\%$ (mean±s.d.) of medium-sized sealing zones ($>500$–$2000$ µm$^2$), $86.3\pm21.\%1$ for larger zones ($>2000$–$5000$ µm$^2$) and $100.0\%$ for very large zones ($>5000$ µm$^2$) exhibited resorption activity. In contrast, AKAP6 depletion resulted in a significant reduction in resorption activity for medium-sized ($>500$–$2000$ µm$^2$) and large ($>2000$–$5000$ µm$^2$) osteoclasts, with only $15.9\pm16.7\%$ and $27.9\pm24.5\%$ of these zones exhibiting resorption activity, respectively. Even very large zones ($>5000$ µm$^2$) in AKAP6-depleted osteoclasts showed reduced resorption efficiency, with only $47.0\pm17.2\%$ exhibiting activity. These data show that the NE-MTOC, regulated by AKAP6 is required for efficient bone resorption by osteoclasts.

## DISCUSSION

We conclude that osteoclasts possess both NE and centrosomal MTOCs, with AKAP6 playing a crucial role in regulating the NE-MTOC without impacting the centrosomal MTOC. This conclusion is supported by several lines of evidence. First, AKAP6 depletion results in the loss of PCNT, CDK5RAP2 and Golgi from the NE without altering their centrosomal localization. In addition, MTOC activity at the NE was specifically inhibited, whereas centrosomal MTOC activity remained unaffected. This distinction enables the differentiation of the functions of these two MTOCs in osteoclasts. Our findings demonstrate that NE-MTOC disruption, through AKAP6 depletion, impairs podosome formation, leading to impaired osteoclast activity.

Although nesprin-1α and AKAP6β were initially identified as muscle-specific isoforms (Duong et al., 2014; Michel et al., 2005), our findings suggest that they are not restricted to muscle tissue. Instead, nesprin-1α and AKAP6β are specifically and robustly induced during the differentiation of RAW264.7-derived osteoclasts. Their expression increased early during osteoclastogenesis (day 1 post-RANKL treatment) and followed a

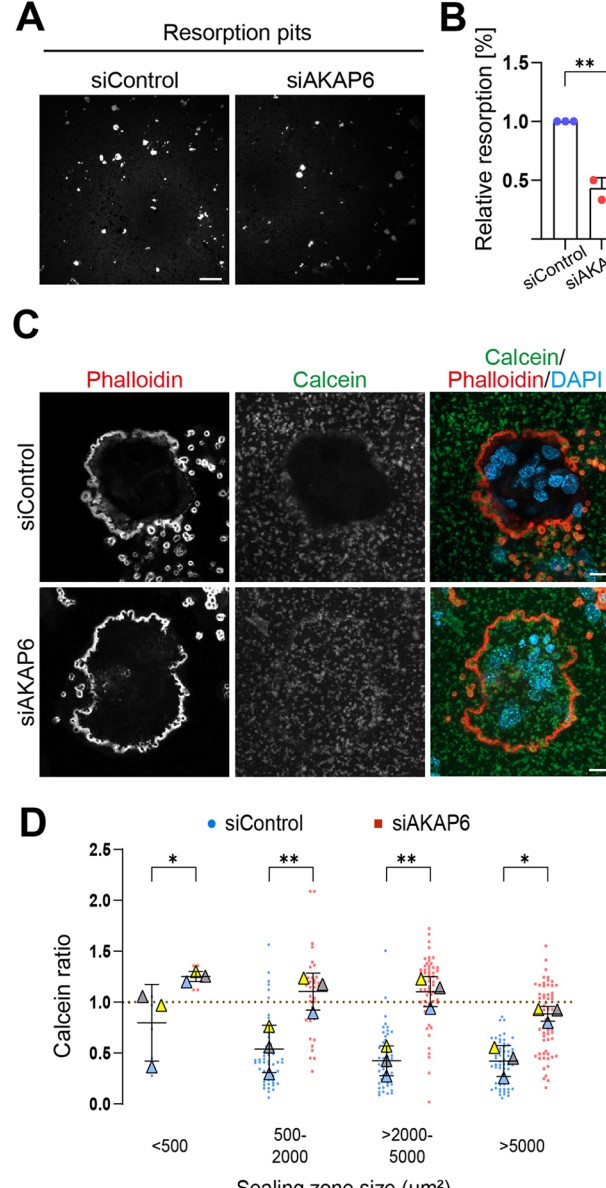

**Fig. 7. AKAP6 depletion impairs osteoclast resorptive activity in RAW264.7-derived osteoclasts.** (A) Representative images of RAW264.7-derived osteoclasts transfected with siControl- and siAKAP6 cultured on CaP-coated wells. Cells were removed, and the CaP substrate was stained with 2.5% AgNO$_3$ to visualize calcium deposits; resorption pits appear as unstained areas. Scale bars: 250 μm. (B) Relative resorbed area of siControl and siAKAP6-treated osteoclasts. Data represent mean±s.d. of three independent experiments, with three wells per experiment. **$P$=0.0081 (paired two-tailed $t$-test). (C) Representative images of RAW264.7-derived osteoclasts transfected with siControl or siAKAP6, and stained for F-actin (phalloidin, red), CaP (calcein, green) and nuclei (DAPI, blue). Note, calcein stains CaP, meaning resorption pits with lower calcein intensity indicate higher osteoclast resorption. Scale bars: 10 μm. (D) Quantification of osteoclast resorptive activity across sealing zone size categories (<500 μm$^2$, 500–2000 μm$^2$, >2000–5000 μm$^2$ and >5000 μm$^2$). SuperPlots display individual resorption pits as color-coded dots (siControl, blue; siAKAP6, red), with biological replicate means ($n$=3; 35–80 pits per experiment, equal numbers per condition) superimposed as colored triangles, each color representing a different replicate. Error bars show overall mean±s.d. Resorption was quantified using the calcein ratio (calcein intensity inside the sealing zone divided by the intensity outside), where values <1 indicate active resorption. *$P$<0.05; **$P$<0.01 (two-way ANOVA with Bonferroni's post-hoc test).

pattern comparable to established osteoclast differentiation markers. Interestingly, all mammalian cells known to assemble a NE-MTOC, skeletal muscle cells, cardiomyocytes and osteoclasts, exhibit a conserved expression pattern of nesprin-1α and AKAP6β isoforms, suggesting a shared mechanism governing NE-MTOC formation across different cellular contexts. Further supporting this hypothesis, the conformational change in SUN2 upon nesprin binding observed during muscle differentiation (Sharma and Hetzer, 2023) is also conserved in osteoclasts. Additionally, we have previously shown that the ectopic expression of nesprin-1α and AKAP6β in non-muscle cells, with an active centrosomal MTOC, is sufficient to induce the recruitment of MTOC proteins to the NE (Vergarajauregui et al., 2020), implying that these isoforms serve as key components of the NE-MTOC.

Although we have demonstrated that AKAP6 depletion specifically affects the NE-MTOC, we cannot fully exclude the possibility that AKAP6 has other signaling roles in osteoclasts. AKAP6 has been implicated in regulating myoblast differentiation (Vargas et al., 2012) and cardiomyocyte hypertrophy (Passariello et al., 2015) by acting as a signaling hub. However, our data argue against a general signaling role for AKAP6 in osteoclast differentiation, as AKAP6 depletion in RAW264.7-derived osteoclasts did not impair their ability to differentiate, as evidenced by the formation of multinucleated cells and the expression of osteoclast-specific differentiation markers analyzed by qPCR (Fig. 4B). These findings suggest that the role of AKAP6 in osteoclasts is specifically tied to NE-MTOC organization rather than broader signaling functions related to differentiation.

A crucial aspect of our study is the ability to distinguish between NE and centrosomal MTOC activity. Unlike pharmacological agents that disrupt the entire MT network, AKAP6 depletion specifically disrupts the NE-MTOC without impairing centrosomal MTOC activity. This specificity allows the assessment of the distinct roles of these MTOCs in osteoclast function. Centrosome-declustering agents (dynarrestin, CW069 and GF) in RAW264.7 cells have been used to conclude that centrosome clustering is essential for osteoclast activation (Philip et al., 2022). However, these agents affect key regulators of MT function, including cytoplasmic dynein and KIFC1, which are also involved in NE-MTOC-mediated processes, such as nuclear alignment in skeletal muscle cells (Espigat-Georger et al., 2016) and MT sliding in neuronal migration (Muralidharan et al., 2022). Our findings reinforce this concern by showing that GF treatment alters not only the organization of centrosomal MTs but also affects the NE-MTOC, as evidenced by disrupted PCNT distribution and shortened MTs concentrated at MTOC sites. This illustrates that declustering drugs might have broader effects than previously appreciated, potentially confounding interpretation when used to assess centrosome-specific functions. In contrast, AKAP6 depletion offers a mechanistically precise strategy to dissect NE-MTOC-dependent pathways. This highlights the distinct and complementary roles of these two MTOCs in osteoclast function and underscores the need for precise experimental approaches to study centrosome clustering. Furthermore, our findings challenge the specificity of conclusions drawn from the use of declustering agents, emphasizing the importance of differentiating between centrosome clustering and overall MT network disruption.

The inactivation of the NE-MTOC through AKAP6 depletion disrupted the organization of the MT network, resulting in MTs failing to reach the F-actin-rich regions at the osteoclast periphery. This suggests a disruption of the critical MT–F-actin crosstalk required for proper cytoskeletal organization and function. Philip et al. reported that centrosome declustering reduces F-actin ring size. The role of MTs in F-actin ring stability is well documented

(Blangy et al., 2020), with complete depolymerization impairing its integrity (Destaing et al., 2005). However, we found that AKAP6 depletion, although affecting podosome maturation, does not alter F-actin ring size on osteoclasts grown on CaP substrates. This discrepancy highlights the importance of distinguishing between centrosome clustering and broader MT network disruption.

Collectively, our findings establish that AKAP6 plays a crucial role in regulating NE-MTOC activity, influencing the MT network and osteoclast resorption function. By specifically disrupting the NE-MTOC without affecting centrosomal MTs, we highlight the distinct roles of these two MTOCs in osteoclast function. Further research is needed to fully elucidate how MTs emanating from different MTOCs differ from each other (e.g. post-translational modifications) and how NE-MTOC inactivation alters MT dynamics, MT–F-actin crosstalk, intracellular trafficking (e.g. lysosomes) and bone resorption. Our study provides a framework for future investigations into the differences between centrosomal and NE-MTOCs, advancing our understanding of osteoclast cytoskeletal organization and activity. In addition, our data suggest AKAP6 as a promising therapeutic target for bone disease. Bone disorders, including osteoporosis, affect millions worldwide, and current treatments often cause severe side effects. Unlike existing approaches, AKAP6 depletion selectively disrupts NE-derived MTs, reducing osteoclast activity without broadly affecting the cytoskeleton. This targeted strategy could improve bone disease treatment while minimizing off-target effects.

## MATERIALS AND METHODS

### Reagents and antibodies
The following primary antibodies were used in this study: rabbit anti-AKAP6 (Sigma-Aldrich, HPA048741; 1:500), mouse anti-GM130 (BD Biosciences, 610823; 1:500), rabbit anti-PCNT (Abcam, ab220784; 1:500), mouse anti-tubulin-γ (Sigma-Aldrich, T6557; 1:200), rabbit anti-CDK5RAP2 (Abcam, ab70213; 1:500), rabbit anti-PCM1 (Proteintech, 19856-1-AP; 1:500), rat anti-tubulin-α (Abcam, ab6160; 1:500), rabbit anti-EB1 (Sigma-Aldrich, E3406; 1:500), mouse anti-nesprin-1 (Developmental Studies Hybridoma Bank, MANNES1E(8C3); 1:200), rabbit anti-SUN2C (Abcam, ab124916; 1:500), and mouse anti-lamin B1 (B-10) (Santa Cruz Biotechnology, sc-374015; 1:500). The following secondary antibodies were used: donkey anti-mouse-IgG Alexa Fluor 647 (1:500, A31571), donkey anti-rabbit-IgG Alexa Fluor 488 (1:500, A21206), donkey anti-rat-IgG Alexa Fluor 594 (1:500, A21209) (all Life Technologies GmH, Darmstadt, Germany). Phalloidin–Alexa Fluor 488 (Thermo Fisher Scientific, A12379; 1:200), phalloidin–Alexa Fluor 594 (Thermo Fisher Scientific, A12381; 1:200), calcein (Roth, 7688.1; 1:500), nocodazole (Sigma-Aldrich, M1404), DTT (Sigma-Aldrich, 646563) and griseofulvin (MedChem Express, HY-17583) were also used.

### Cell culture
RAW264.7 cells were obtained from ATCC and maintained in high glucose DMEM supplemented with GlutaMAX (Thermo Fisher Scientific, 61965059) containing 10% heat-inactivated fetal bovine serum (FBS; Gibco, 26140079), 1 mM sodium pyruvate, 100 U/ml penicillin and 100 µg/ml streptomycin (Thermo Fisher Scientific, 15140-122) and cultured at 37°C in a humidified atmosphere containing 5% $CO_2$. Cells were authenticated based on their ability to robustly differentiate into multinucleated osteoclasts in response to RANKL and were routinely tested for mycoplasma contamination (every 12 months). For differentiation, cells were plated at 20,000 cells per ml in αMEM (Thermo Fisher Scientific, 41061029) supplemented with 10% heat-inactivated FBS and 50 ng/ml RANK-L (R&D Systems, 462-TEC-010). Medium was replaced every 2 days with large multinucleated osteoclasts formed at day 4.

### siRNA transfection
RAW264.7 cells were transfected on day 1 of differentiation with siRNA control (Silencer Select Negative Control No.1) or siRNA against AKAP6 (Silencer Select s108732) at 20 nM final concentration by addition of transfection complexes pre-formed for 20 min, containing 0.3 ml Lipofectamine RNAiMAX (Life Technologies) per pmol of siRNA in Opti-MEM medium. Cells were processed 3 days after transfection. A medium change with RANKL was performed 24 h after addition of the siRNA mixtures to the RAW264.7 cells.

### RNA isolation, reverse transcription and PCR
RNA was isolated from RAW264.7 cells at specific differentiation stages using Macherey-Nagel's Nucleospin RNA Plus isolation kit (Cat# 740984.250), followed by reverse transcription utilizing 1 µg of RNA, random primers and M-MLV Reverse Transcriptase (Sigma-Aldrich, Cat# M1302-40KU). qPCR was performed on a CFX Connect Real-Time PCR system (Bio-Rad) using 10 ng cDNA, SYBR Green Master Mix (Life Technologies) and gene-specific primers in a 10 µl reaction. Cycling conditions were: 98°C for 30 s, followed by 40 cycles of 98°C for 15 s and 60°C for 30 s. Each sample was run in technical triplicate. Mean Cq values were normalized to *Gapdh*, and relative transcript levels were calculated using the ΔΔCt method ($2^{-\Delta\Delta Ct}$). The following primers were used: *Syne1giant*: forward 5′-ACCTCATGGCAGCTAGGGTG-3′, reverse 5′-TATGGGCTTGGCCAACTCTG-3′; *Syne1a*, forward 5′-ACAGACA-GACAACCAGCACC-3′, and reverse, 5′-TATGGGCTTGGCCAACTC-TG-3′; *Akap6*, forward, 5′-TCTGGGGACATAAGTGTGAG-3′, and reverse, 5′-CCTGAATGATGCGTTGGACT-3′; *Akap6a*, forward, 5′-ACATGACACCTACCGTGGAG-3′, and reverse, 5′-CCGAGTAGGT-TAGGTCACGG-3′; *Akap6b*, forward, 5′-TCTAAAGCAGTTAGGCCCA-CAG-3′, and reverse, 5′-CTTCGGATGAGCTCGGGAAAT-3′; *Gapdh*, forward, 5′-CAGAAGACTGTGGATGGCCC-3′, and reverse, 5′-AGTG-TAGCCCAGGATGCCCT-3′; *Emr1*, forward, 5′-TTGTACGTGCAACT-CAGGACT-3′, and reverse, 5′-GATCCCAGAGTGTTGATGCAA-3′; *Trap*, forward, 5′-ACTTCCCCAGCCCTTACTACCG-3′, and reverse, 5′-TCAGCACATAGCCCACACCG-3′; *Ctsk*, forward, 5′-AGGCGGCTA-TATGACCACTG-3′, and reverse, 5′-CCGAGCCAAGAGAGCATATC-3′; *Nfatc1*, forward, 5′-TGCTCCTCCTCCTGCTGCTC-3′, and reverse, 5′-CGTCTTCCACCTCCACGTCG-3′. All primers were from Sigma-Aldrich.

### CaP coating
CaP coatings were prepared following a modified protocol from Maria et al. (2014). Coverslips in 24-well culture plates were incubated with 600 µl of freshly made simulated body fluid (SBF) for 3 days at room temperature (RT). SBF was generated by mixing 50% Tris buffer (50 mM Tris pH 7.4), 25% calcium stock solution (25 mM $CaCl_2 \cdot H_2O$, 1.37 M NaCl, 15 mM $MgCl_2 \cdot 6H_2O$ in Tris buffer, pH 7.4), and 25% phosphate stock solution (11.1 mM $Na_2HPO_4 \cdot H_2O$, 42 mM $NaHCO_3$ in Tris buffer, pH 7.4). Then the SBF was replaced with fresh 600 µl of CaP solution (CPS) for 24 h. CPS was prepared by dissolving 2.25 mM $Na_2HPO_4 \cdot H_2O$, 4 mM $CaCl_2 \cdot 2H_2O$, 0.14 M NaCl and 50 mM Tris in MilliQ water containing 41 ml 1 M HCl, adjusting the pH to 7.4, and bringing the final volume to 1 l. Following incubation, the CPS was aspirated, and coated coverslips were dried under nitrogen flow. The plates were sterilized under UV light for 30 min before use and either used immediately or stored dry at RT for up to 1 month. Prior to cell seeding, coated coverslips were incubated with FBS for 1 h at 37°C.

### Characterization of osteoclastic resorption
To assess the resorption activity of RAW 264.7-derived osteoclasts, cells were differentiated on CaP-coated coverslips for 5 days. Resorption was visualized with a von Kossa staining. In brief, cells were lysed with water and incubated with 2.5% silver nitrate under UV light until dark brown or black stain develops, then washed in distilled water three times. The surface of the CaP was examined by bright field microscopy and resorption areas were measured using ImageJ. To enable simultaneous visualization of sealing zones and resorption pits by fluorescence microscopy, we adapted the protocol described by Cen et al. (2022). Cells were differentiated on CaP-coated coverslips for 5 days, fixed with 4% PFA for 15 min, permeabilized with 0.1% Triton X-100 in PBS for 5 min, and washed in PBS. F-actin was stained with phalloidin–Alexa Fluor 594 (1:300, 1 h), nuclei with DAPI (0.5 µg/ml, 10 min) and CaP coating with calcein (10 µM, 30 min). All steps

were performed at room temperature with PBS washes between steps. Confocal imaging was used to visualize sealing zones (phalloidin) and remaining CaP (calcein). Regions of interest (ROIs) were defined by the phalloidin-positive sealing zones, and calcein intensity was measured within each ROI. For normalization, calcein intensity was also measured in an equivalent area immediately outside the sealing zone. Resorption activity was expressed as the ratio of calcein intensity inside versus outside the sealing zone, with ratios <1 indicating local CaP degradation.

### MT regrowth assay

RAW264.7-derived osteoclasts were treated with 10 mg/ml nocodazole (Sigma-Aldrich) for 1 h in the corresponding differentiation medium. After three washes with cold PBS, cells were let to recover in fresh differentiation medium at 37°C in 5% $CO_2$. Cells were then fixed with methanol and stained as indicated below.

### Immunofluorescence

Cells were fixed either in ice-cold methanol for 5 min at −20°C or in 4% PFA for 15 min RT and then permeabilized with 0.5% Triton X-100 in PBS for 5 min at RT. Cells were blocked in 10% FBS and 0.1% saponin in PBS (blocking buffer) for 30 min at RT. Primary antibodies were incubated in blocking buffer for 1–2 h at RT or at 4°C overnight. Following primary antibody incubation, cells were washed with PBS, incubated with fluorophore-conjugated secondary antibodies in blocking buffer for 1 h at RT and washed with PBS. DNA was visualized with 0.5 mg/ml DAPI (Sigma) in 0.1% NP40 in PBS. After DAPI staining, coverslips were rinsed once with Millipore-filtered water and then mounted using Fluoromount-G mounting medium (Life Technologies GmbH). Analysis, image acquisition, and high-resolution microscopy were done using a LSM800 confocal laser scanning microscope equipped with an Airyscan detector and the ZEISS Zen (Blue edition) software with Airyscan image processing.

### Image preparation and analysis

Images were prepared by using ImageJ and Adobe Photoshop. All images were modified by adjustments of levels and contrast. ImageJ was used for quantification of the immunofluorescence signal intensity. To measure the distribution of immunofluorescence signal intensity of Golgi or MTs around the NE along the cell, we selected each nuclei of the osteoclasts as a ROI. Like this, the differences in nuclei size and/or circularity were avoided, and distances from the nuclear edge were more precise. Every nuclear ROI was increased in 0.2 μm steps (−2 to +2 μm). At each step a 0.2 μm wide band was created and the fluorescence mean intensity from each band was measured. The fluorescence intensity values for each band were normalized to the mean intensity of the entire −2 to +2 μm region. For each nucleus, the position of the maximum intensity (peak position) and the maximum intensity value itself (peak intensity) were extracted from the intensity profile.

For quantifying actin–MT crosstalk at the podosome belt, high-resolution images of podosome belt segments were acquired. The belt region contained within each image was manually delineated using the polygon tool in ImageJ, based on the continuous F-actin ring. Colocalization analysis was performed using the BIOP JACoP plugin (Bolte and Cordelieres, 2006), a toolbox that computes pixel-based colocalization metrics (including Manders' coefficients, Pearson's correlation coefficient, and overlap fractions) between two fluorescence channels. For this study, we used the pixel overlap measurement (Manders' coefficient M1) to quantify the fraction of the F-actin signal that colocalized with MTs. Colocalization (overlap) values were expressed as arbitrary units (a.u.) representing the proportion of F-actin pixels overlapping with MT pixels relative to the total F-actin signal.

### Statistical analysis

Statistical comparisons and graph production were performed in Graph Prism using a two-tailed unpaired $t$-test, a two-tailed paired $t$-test or two-way ANOVA with the post-hoc test Bonferroni. Three biological replicates were performed per experiment. Biological replicate means independent RAW264.7 differentiations, treated, fixed and stained, and analyzed. To appropriately account for hierarchical data structure, superplots were used for all quantifications; individual data points are shown, color-coded by biological replicate, and statistical testing was performed on the replicate-level means rather than on individual cells.

### Acknowledgements

We thank Antonio Maccataio for technical assistance.

### Competing interests

The authors declare no competing or financial interests.

### Author contributions

Conceptualization: S.V., F.B.E.; Data curation: S.V., S.P.; Formal analysis: S.V., S.P.; Funding acquisition: U.S., F.B.E.; Investigation: S.V., S.P., J.O.O.; Methodology: S.V., S.P., J.O.O., U.S.; Project administration: S.V., F.B.E.; Resources: U.S.; Supervision: S.V., F.B.E.; Validation: S.V.; Visualization: S.V., S.P., F.B.E.; Writing – original draft: S.V., F.B.E.; Writing – review & editing: S.V., F.B.E.

### Funding

This work was supported by the Deutsche Forschungsgemeinschaft (DFG, INST 410/91–1 FUGG and EN 453/12-3 to F.B.E., and 501752319-TRR369 DIONE project B02 to U.S.), the University Hospital Erlangen (IZKF, project A99 to U.S.) and the Bavarian Californian Technology Center (BaCaTeC, project 17 2021-2 to F.B.E.). Open Access funding provided by University Library Erlangen-Nuremberg (UB). Deposited in PMC for immediate release.

### Data and resource availability

All relevant data and details of resources can be found within the article and its supplementary information.

### Peer review history

The peer review history is available online at https://journals.biologists.com/jcs/lookup/doi/10.1242/jcs.264166.reviewer-comments.pdf

### Special Issue

This article is part of the Special Issue 'Cell Biology of the Nucleus', guest edited by Abby Buchwalter. See related articles at https://journals.biologists.com/jcs/issue/139/12.

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
