## [Peer Review File · Journal of Cell Science]

Selective disruption of microtubule formation at the nuclear envelope impairs the bone resorption capacity of osteoclasts

Silvia Vergarajauregui, Samantha Panea, Jakob O. Oltmanns, Ulrike Steffen and Felix B. Engel

DOI: 10.1242/jcs.264166

Editor: Megan King

Review timeline

Original submission:	26 May 2025
Editorial decision:	16 July 2025
First revision received:	19 November 2025
Accepted:	23 November 2025

Original submission

First decision letter

MS ID#: jcs.264166

MS TITLE: Selective disruption of microtubule formation at the nuclear envelope impairs the bone resorption capacity of osteoclasts

AUTHORS: Silvia Vergarajauregui; Samantha Panea; Jakob O Oltmanns; Ulrike Steffen; Felix B Engel
ARTICLE TYPE: Research Article

Dear Dr Vergarajauregui,

We have now reached a decision on the above manuscript.

To see the reviewers' reports and a copy of this decision letter, please go to:

As you will see, while Reviewer 1 was quite enthusiastic about the manuscript and offered minor suggestions to improve the paper, Reviewer 2 felt that your prior published work diminishes the conceptual advance and impact of this study. I would like to provide you with the opportunity to address this critique. If you think that you can deal satisfactorily with the criticisms on revision, I would be pleased to see a revised manuscript.

Reviewer 1

Advance summary and potential significance to field

In the manuscript entitled "Selective disruption of microtubule formation at the nuclear envelope impairs the bone resorption capacity of osteoclasts," the authors investigate the role of microtubule-organizing centers (MTOCs) in osteoclasts. The manuscript reveals that osteoclasts possess both centrosomal and nuclear envelope (NE)-associated MTOCs. It identifies AKAP6¹ and nesprin-1a as key proteins for NE-MTOC formation, showing their upregulation during osteoclast differentiation and suggesting a conserved role across cell types. It describes that targeted depletion of AKAP6 selectively impairs microtubule nucleation from the NE, displacing associated proteins and disrupting Golgi organization, without affecting centrosomal MTOCs. This specific

disruption of NE-MTOC activity leads to impaired podosome formation and a significant reduction in bone resorption capacity, underscoring the distinct and essential function of NE-derived microtubules in osteoclast activity. The manuscript is well written and the results appropriate for publication in the Journal of Cell Science, upon revision of the multiple points mentioned below:

Line 143 - the authors should describe here their model system (RAW264.7-derived osteoclasts after 4 days of RANKL addition) as they do in the figure 1 legend.

Fig 1 - The authors effectively demonstrate that osteoclasts, akin to skeletal and cardiac muscle cells, recruit centrosomal proteins such as AKAP6, PCNT, CDK5RAP2, and PCM1 to the nuclear envelope, which subsequently acts as a non-centrosomal microtubule-organizing center. A notable observation appears to be the differential recruitment of PCNT and CDK5RAP2 to the NE of nuclei within the same cell, while AKAP6 and PCM1 seem to be uniformly distributed across all nuclei. If this differential pattern holds true, a discussion on this finding would be required.

Line 165 - The statement regarding 'similar characteristics' on Line 165 is rather broad. It is suggested to narrow this statement, as the authors primarily show that the NE in osteoclasts, similar to muscle cells, contains PCM1, PCNT, AKAP6, and CDK5RAP2.

Fig 2B - Quantification of the number of nuclei positive for EB1 one minute after regrowth in Figure 2B should be performed.

Figure 2C- Quantifying the number of nuclei exhibiting microtubule regrowth in both GM130 positive and negative nuclei would provide valuable insights.

Line 210: This section begins by detailing the known roles of AKAP6 and nesprin as MTOC anchors in cardiac and skeletal muscle, subsequently transitioning into their characterization in osteoclasts (Figure 3). Given that AKAP6's presence at the NE in osteoclasts has already been established in Figures 1 and 2, it is suggested that this section primarily focus on the characterization of nesprin-1, with less emphasis on AKAP6, to streamline the narrative and avoid redundancy.

Line 221: Please correct the typo in "immunostaining" on Line 221.

Line 226 - Figure 3A and 3B Integration: To better integrate with the RT-PCR data presented in Figure 3B, it is crucial for the authors to quantify the number of undifferentiated RAW264.7 cells in their assays and provide supportive data confirming their undifferentiated state. Additionally, for Figure 3A, displaying independent panels for undifferentiated and differentiated osteoclasts, clearly stained for AKAP6 and Nesprin-1, would more effectively illustrate the absence of AKAP6 and Nesprin-1 at the nuclear envelope in undifferentiated cells.

Figure 3B - RT-PCR Quantification: While RT-PCR was used in Figure 3B to characterize nesprin and AKAP6 isoform levels during differentiation, for enhanced quantitative rigor, it is recommended that the authors consider performing real-time PCR, if feasible, to strengthen their findings.

Figure 4 - Given that Figure 4 presents siRNA experimental results, quantifying all observed phenotypes in this figure is important to assess the penetrance of the phenotype and provide a more robust analysis.

Figure 5A: The authors assert that AKAP6 knockdown leads to a specific loss of microtubule outgrowth from the NE, but not from centrosomes, thus selectively impairing NE-MTOC activity. However, it appears that AKAP6 depletion might also affect non-nuclear MTOCs, as indicated by the alpha-tubulin staining from PCNT spots outside the nucleus not exhibiting the characteristic astral nucleation seen in the control. The authors should investigate if microtubule nucleation is also impaired from centrosomal MTOCs in AKAP6-depleted cells using the nocodazole release assay, to evaluate the selectivity of the impairment.

Figure 6A: the inclusion of the BIOP JACoP channel below the individual cytoskeleton panels (actin and tubulin) on the top and middle seems less informative. displaying the bottom panel, which illustrates the co-localization of actin and tubulin, provides a more direct and valuable visual representation.

Line 374: it is suggested that the authors rephrase 'released' to 'not localized to the nuclear envelope' when referring to PCNT and CDKRAP2, as this phrasing more accurately reflect their observation.

Reviewer 2

Advance summary and potential significance to field

In the study entitled "Selective disruption of microtubule formation at the nuclear envelope impairs the bone resorption capacity of osteoclasts", Silvia Vergarajauregui and colleagues explore microtubule nucleation at the nuclear envelopes in osteoclasts and its implication in the resorption function of osteoclasts.

The present study builds on a previous report by the same research team entitled "AKAP6 orchestrates the nuclear envelope microtubule-organizing center by linking golgi and nucleus via AKAP9" published in eLife in 2020, showing that the nuclear envelopes of osteoclasts nucleate microtubules in an AKAP6-dependent fashion and that AKAP6 is required for osteoclast resorption function; that study was performed in osteoclast derived from peripheral blood monocytes (10.7554/eLife.61669).

Here, the authors used the RAW264.7 mouse macrophage-like cell line, a widely used model to differentiate osteoclasts.

Here the authors claim that in RAW264.7-derived osteoclasts:

- MTOC and Golgi proteins at the nuclear envelopes (Figure 1)
- the nuclear envelopes, associated or not with Golgi, nucleate microtubules (Figure 2)
- express different isoforms of Nesprin1 and AKAP6 (Figure 3)
- that AKAP6 is required for recruitment of Golgi and MTOC components at the nuclear envelope, and for the nuclear envelope to organize microtubules (Figure 4)
- but that AKAP6 is not required for centrosomal microtubule nucleation (Figure 5)
- a reduced in the overlap between actin and microtubules at the actin ring in the absence of AKAP6 (Figure 6)
- reduced resorption activity in the absence of AKAP6 (Figure 7)

General comment:

Overall, the images are of good quality and the manuscript is clearly written.

Nevertheless, the present study recapitulates the former findings in human osteoclasts, without providing any new mechanistic insights into the link between microtubule nucleation at the nuclear envelope and the osteoclasts resorption function.

Several experimental approaches lack sufficient description, in particular the method of quantification by RT-PCR, the quantification of images using BIOP JACoP and the method to quantify the resorption activity, the quantifications in figures 6 and 7 lack robustness.

Comments for the author

Major comments

- line 205-206: it is does not seem appropriate to conclude "These results suggest that pharmacological disruption of the clustering of centrosomes affects both centrosomal and nuclear envelope-associated MTOCs.", because Griseofulvin binds to tubulin and interferes with microtubule dynamics; thus, it is not possible to link Griseofulvin effects on microtubule nucleation to a defect in centrosome clustering.
- Figure 4A/lines 266-267: how was PCR performed to be able to quantify Akap6 mRNA? No protocol is specified in the methods section, and the results shown are end point PCR, not Q-PCR.
- Figure 4A: comment on the fact that Syne1 giant mRNA is not detected here in osteoclast, as compared to the kinetics shown in 3B.
- Figure 5B: how was significance of the intensity differences assessed?
- Figure 5D: The increase appears around 30%, which is not "slight" (line 308).
- Figure 6: what is the BIOP JACoP macro, how was it used? How were ROI chosen? How many experiments were performed? Figure 6B suggests it is only 1 experiment counting around 30 cells / ROI, in any case either it contradicts what is stated in the methods section lines 560-561 "All experiments were repeated at least three times.", otherwise the p value would not be <0.0001. Of note, t-test is not appropriate for few experimental repeats, non-parametric test should be performed. The experimental approach must be clarified. What does "displayed less-defined F-actin rings" mean line 341, how was it assessed? Figure 6A does not suggest a less defined F-actin ring in the siAKAP6, neither does Figure 7A.
- Figure 7: same comments as above about experiment repetition and statistical analyses: the stats in 7B graph suggests that one cells was counted 1 experimental repetition. This has to be

explained. The quantification approach is odd. The methods section states lines 528-530: "The ratio of calcein intensity inside the sealing zone to that outside was calculated. A ratio below 1 indicated CaP resorption." How can the authors explain that the calcein ratio over 1 in cells with the smallest sealing zones and in all siAKAP6 sample? The mean/median should have a maximum around 1, not around 2; otherwise it suggests that osteoclasts accumulate calcium phosphate within the sealing zones. How/where was outside calcein intensity measured? The authors refer to Cen et al., 2022 for quantification, but the quantification approach is not as in that paper, which uses the classical method to measure the surface resorbed in the whole well, not below selected cells.

- About image quantification, lines 559-560: "The fluorescence intensity values for each band were normalized to the mean intensity of the whole cell." How was total fluorescence measured in osteoclast for normalization, provided that no entire cell is imaged in the present study.

First revision

Author response to reviewers' comments

Reviewer 1: In the manuscript entitled "Selective disruption of microtubule formation at the nuclear envelope impairs the bone resorption capacity of osteoclasts," the authors investigate the role of microtubule-organizing centers (MTOCs) in osteoclasts. The manuscript reveals that osteoclasts possess both centrosomal and nuclear envelope (NE)-associated MTOCs. It identifies AKAP6B and nesprin-1a as key proteins for NE-MTOC formation, showing their upregulation during osteoclast differentiation and suggesting a conserved role across cell types. It describes that targeted depletion of AKAP6 selectively impairs microtubule nucleation from the NE, displacing associated proteins and disrupting Golgi organization, without affecting centrosomal MTOCs. This specific disruption of NE-MTOC activity leads to impaired podosome formation and a significant reduction in bone resorption capacity, underscoring the distinct and essential function of NE-derived microtubules in osteoclast activity. The manuscript is well written and the results appropriate for publication in the Journal of Cell Science, upon revision of the multiple points mentioned below:

We thank the reviewer for the thorough and constructive evaluation of our manuscript. We appreciate the insightful comments and have carefully addressed each point as detailed below.

All suggested changes have been incorporated into the revised manuscript, and specific line numbers refer to the changes, which are also highlighted in yellow.

Line 143 - the authors should describe here their model system (RAW264.7-derived osteoclasts after 4 days of RANKL addition) as they do in the figure 1 legend.

We thank the reviewer for this suggestion. We have updated Lines 145-146 to include a clear description of our model system, consistent with the legend of Figure 1. The text now reads: "...To address this controversy, the MTOCs from RAW264.7-derived osteoclasts after 4 days of RANKL addition were characterized."

Fig 1 - The authors effectively demonstrate that osteoclasts, akin to skeletal and cardiac muscle cells, recruit centrosomal proteins such as AKAP6, PCNT, CDK5RAP2, and PCM1 to the nuclear envelope, which subsequently acts as a non-centrosomal microtubule-organizing center. A notable observation appears to be the differential recruitment of PCNT and CDK5RAP2 to the NE of nuclei within the same cell, while AKAP6 and PCM1 seem to be uniformly distributed across all nuclei. If this differential pattern holds true, a discussion on this finding would be required.

We thank the reviewer for this perceptive comment. In our previous work (Das et al., 2025), we demonstrated that the recruitment of centrosomal proteins to the nuclear envelope in differentiating myoblasts occurs in a gradual and sequential manner. Given that osteoclasts are formed by the fusion of multiple precursor cells, and that RAW264.7-derived osteoclasts in particular can be very large, we propose that NE-MTOC assembly is not synchronized across all nuclei. To address this point, we have now quantified the proportion of nuclei positive for each centrosomal protein. This analysis shows that while AKAP6 and PCM1 are present at the nuclear envelope of nearly all nuclei, only ~82% of nuclei show enrichment of PCNT and CDK5RAP2. This variability may reflect differences in nuclear maturation and/or the stage of NE-MTOC

establishment within individual nuclei. We have now incorporated the quantification data of this heterogeneous localization pattern, which may reflect asynchronous or stage-specific recruitment of certain centrosomal components during osteoclast maturation.

To address this point, we have added this explanation to the Results (Lines 163-173) which reads: “Notably, while AKAP6 and PCM1 are present at the nuclear envelope of nearly all nuclei, the recruitment of PCNT and CDK5RAP2 is more heterogeneous. Only $82.33 \pm 13.5 \%$ or $82.67 \pm 4.0 \%$ of nuclei show enrichment of CDK5RAP2 or PCNT, respectively. We recently demonstrated that the recruitment of centrosomal proteins to the nuclear envelope in differentiating myoblasts occurs in a gradual and sequential manner (Das et al., 2025). Similarly, the differential recruitment of centrosomal proteins to the NE in RAW264.7-derived osteoclasts may reflect variable stages of nuclear maturation and/or progressive establishment of NE-MTOC within individual nuclei.”

Line 165 - The statement regarding 'similar characteristics' on Line 165 is rather broad. It is suggested to narrow this statement, as the authors primarily show that the NE in osteoclasts, similar to muscle cells, contains PCM1, PCNT, AKAP6, and CDK5RAP2.

We agree with the reviewer that the original statement was too broad. We have now revised this sentence: “Collectively, these findings suggest that the organization of the NE-MTOC in RAW264.7-derived osteoclasts follows the same pattern as that in skeletal muscle cells and cardiomyocytes based on the presence of MTOC proteins.” Lines 174-176.

Fig 2B - Quantification of the number of nuclei positive for EB1 one minute after regrowth in Figure 2B should be performed.

We agree with the reviewer and have now quantified the percentage of nuclei exhibiting EB1 puncta 1 minute after nocodazole washout. We find that $94 \pm 2.5\%$ of nuclei were EB1-positive, confirming robust microtubule regrowth from the nuclear envelope. These results have been added to the Results section (Line 190).

Figure 2C- Quantifying the number of nuclei exhibiting microtubule regrowth in both GM130 positive and negative nuclei would provide valuable insights.

As suggested, we have now quantified microtubule regrowth in nuclei that are either GM130-positive or GM130-negative. This analysis is included in the revised Results section (Line 202) and reads: “while most of the nuclei were positive for both PCNT and GM130 ($85 \pm 0.09 \%$), we observed some nuclei ($12 \pm 0.04 \%$) which exhibited MT outgrowth were positive for PCNT but lacked GM130.” (Line 201-203)

Line 210: This section begins by detailing the known roles of AKAP6 and nesprin as MTOC anchors in cardiac and skeletal muscle, subsequently transitioning into their characterization in osteoclasts (Figure 3). Given that AKAP6's presence at the NE in osteoclasts has already been established in Figures 1 and 2, it is suggested that this section primarily focus on the characterization of nesprin-1, with less emphasis on AKAP6, to streamline the narrative and avoid redundancy.

We thank the reviewer for this insightful comment. We agree that emphasizing nesprin-1 α improves the clarity and flow of the section, and we have revised the text accordingly (Page 7, lines 228-256) so that the narrative is primarily centered on the characterization of nesprin-1 α during osteoclast differentiation.

At the same time, we have retained the analysis of Akap6 transcript regulation in Fig. 3 for two reasons:

- This information has not been shown in Figures 1 or 2, which focus exclusively on AKAP6 protein localization, not on its transcriptional induction or isoform-specific regulation during osteoclastogenesis.
- The finding that Akap6b is strongly and early upregulated, mirroring Syne1a induction, is novel and essential to the conclusion that osteoclasts activate the same nesprin-1 α /AKAP6B module previously considered muscle-restricted. Because the NE-MTOC is built by both proteins, documenting the coordinated induction of their transcripts provides mechanistic insight that cannot be inferred from localization alone.

We believe this revision maintains the reviewer's request for a clearer nesprin-centered narrative while ensuring that the novel and essential transcriptional regulation of Akap6 is not omitted.

Line 221: Please correct the typo in "immunostaining" on Line 221.
This typographical error has been corrected.

Line 226 - Figure 3A and 3B Integration: To better integrate with the RT-PCR data presented in Figure 3B, it is crucial for the authors to quantify the number of undifferentiated RAW264.7 cells in their assays and provide supportive data confirming their undifferentiated state.

We thank the reviewer for this constructive comment. In the revised version of Figure 3, we now present qPCR data (instead of endpoint RT-PCR) to more accurately monitor transcript dynamics during osteoclastogenesis.

Regarding the number of undifferentiated RAW264.7 cells: Direct quantification of "undifferentiated RAW264.7 cells" within the cultures is not feasible in a meaningful or reproducible way. Undifferentiated RAW264.7 cells are very small, mononuclear, and typically appear in tight clumps, whereas RAW264.7-derived osteoclasts are very large multinucleated cells. Because these populations differ dramatically in size, morphology, and spatial organization, they cannot be accurately enumerated within the same field of view or reliably segmented for cell counting.

To address the reviewer's underlying concern, we included molecular markers that unambiguously reflect differentiation status:

- *Emr1*, a macrophage marker, decreases progressively after RANKL stimulation, confirming the loss of the undifferentiated macrophage state.
- *Trap* and *Ctsk*, established osteoclast markers, show strong induction, validating osteoclast differentiation.

This has been included in Figure 3B as a heatmap of qPCR-derived RNA expression levels for *Emr1*, *Akap6*, *Syne1a*, *Trap* and *Ctsk* during osteoclast differentiation (days 1-4 of RANKL treatment) relative to non-differentiated precursor cells, and accordingly in the results section in lines 242-245.

Additionally, for Figure 3A, displaying independent panels for undifferentiated and differentiated osteoclasts, clearly stained for AKAP6 and Nesprin-1, would more effectively illustrate the absence of AKAP6 and Nesprin-1 at the nuclear envelope in undifferentiated cells.

We thank the reviewer for this helpful suggestion. In the revised Figure 3A, we now present independent panels showing representative images of undifferentiated RAW264.7 cells and RANKL-differentiated osteoclasts, each stained for AKAP6 and nesprin-1. These new panels more clearly highlight the absence of both proteins in undifferentiated cells and their robust enrichment at the nuclear envelope in multinucleated osteoclasts. This modification improves the clarity and interpretability of the figure as recommended, and has been added to the Results section (lines 237-242)

Figure 3B - RT-PCR Quantification: While RT-PCR was used in Figure 3B to characterize nesprin and AKAP6 isoform levels during differentiation, for enhanced quantitative rigor, it is recommended that the authors consider performing real-time PCR, if feasible, to strengthen their findings.

We thank the reviewer for this valuable suggestion. In response, we have replaced the previous endpoint RT-PCR analysis with quantitative real-time PCR (qPCR). The updated qPCR data are now presented in three dedicated panels:

- Fig. 3B: qPCR analysis of *Syne1a* (nesprin-1 α) and *Akap6* transcript levels alongside the macrophage marker (*Emr1*) and established osteoclast markers (*Trap*, *Ctsk*), demonstrating coordinated upregulation during differentiation.
- Fig. 3C: Isoform-specific qPCR of *Syne1a* and *Syne1-giant*, showing selective induction of the nesprin-1 α isoform.
- Fig. 3D: Isoform-specific qPCR of *Akap6a* and *Akap6b*, revealing preferential early induction of the muscle-associated AKAP6B isoform during osteoclastogenesis.

These qPCR data provide enhanced quantitative rigor and more clearly support the conclusion that nesprin-1 α and AKAP6B are specifically induced during osteoclastogenesis (Lines 242-256). The methodological details for qPCR have been added to the Materials and Methods section (Lines 528-533).

Figure 4 - Given that Figure 4 presents siRNA experimental results, quantifying all observed phenotypes in this figure is important to assess the penetrance of the phenotype and provide a

more robust analysis.

We thank the reviewer for this important comment. We have now quantified all relevant phenotypes observed in the siRNA experiments: Golgi dispersion and nuclear envelope localization of MTOC-associated proteins in control versus AKAP6-depleted cells. These quantifications are now fully integrated into the Results text to provide a rigorous assessment of phenotype penetrance (Lines 283-296).

Although we agree that graphical representation can be helpful, adding additional quantification panels to Figure 4 was not feasible without compromising figure clarity and layout. Therefore, we have reported instead all quantified values in the text.

Figure 5A: The authors assert that AKAP6 knockdown leads to a specific loss of microtubule outgrowth from the NE, but not from centrosomes, thus selectively impairing NE-MTOC activity. However, it appears that AKAP6 depletion might also affect non-nuclear MTOCs, as indicated by the alpha-tubulin staining from PCNT spots outside the nucleus not exhibiting the characteristic astral nucleation seen in the control. The authors should investigate if microtubule nucleation is also impaired from centrosomal MTOCs in AKAP6-depleted cells using the nocodazole release assay, to evaluate the selectivity of the impairment.

We agree with the reviewer that the previously presented examples gave the impression that there might be some reduction in the centrosomal MTOC activity upon AKAP6 depletion. However, we would like to note that osteoclasts contain different configurations of centrioles (single or multiple centrioles) which exhibit different MTOC activities. These activities appear also to be dependent on the localization (e.g., between nuclei, free in the cytoplasm, or associated to a nucleus at the border of nuclei accumulations). Thus, a comparison is not trivial. In addition, small variations in the timing of the re-growth due to experimental constraints can affect aster length.

To address this point, we have carefully re-assessed all our images and verified that there is no obvious effect of AKAP6 depletion on centrosomal MTOC activity. To clarify this point, we have revised Figure 5C by replacing the original images with higher-quality stainings showing α -tubulin together with EB1 and γ -tubulin following a 1-minute nocodazole washout. In these new images, centrosomal microtubule nucleation is clearly visible in both control and AKAP6-depleted cells, whereas NE-associated nucleation is abolished upon AKAP6 depletion. We now also include zoomed insets highlighting centrosomal MTOCs of different centriole configurations.

Regarding quantification, we have now added to the Results section (lines 331-336) the limitation that aster length cannot be reliably measured (e.g., due to spatial constraints such as location between nuclei or freely in the cytoplasm) and clarified the issue of multiple centriole configurations. Collectively, the now presented immunofluorescence data together with the quantification indicates that AKAP6 depletion has no obvious effect on centrosomal MTOC activity.

Note, based on a comment of reviewer 2 we have revised the statistical analysis of the quantification of centrosomal MTOCs. The new statistical analysis (biological means) shows that there is no significant difference in centrosomal MTOC activity in WT and AKAP6-depleted cells.

Figure 6A: the inclusion of the BIOP JACoP channel below the individual cytoskeleton panels (actin and tubulin) on the top and middle seems less informative. displaying the bottom panel, which illustrates the co-localization of actin and tubulin, provides a more direct and valuable visual representation.

We agree that the current presentation of the top and middle panels is less informative. In the revised figure, we now omit the separate actin and tubulin BIOP channels and retain only the merged co-localization panel at the bottom, which clearly shows the overlap between actin and tubulin. This change improves figure clarity and interpretability.

Line 374: it is suggested that the authors rephrase 'released' to 'not localized to the nuclear envelope' when referring to PCNT and CDKRAP2, as this phrasing more accurately reflect their observation.

We thank the reviewer for pointing this out. We have revised the sentence to state: "..., AKAP6 depletion results in the loss of PCNT and CDK5RAP2, and Golgi from the NE without altering their centrosomal localization..." (Line 412)

Reviewer 2: In the study entitled "Selective disruption of microtubule formation at the nuclear

envelope impairs the bone resorption capacity of osteoclasts", Silvia Vergarajauregui and colleagues explore microtubule nucleation at the nuclear envelopes in osteoclasts and its implication in the resorption function of osteoclasts.

The present study builds on a previous report by the same research team entitled "AKAP6 orchestrates the nuclear envelope microtubule-organizing center by linking golgi and nucleus via AKAP9" published in eLife in 2020, showing that the nuclear envelopes of osteoclasts nucleate microtubules in an AKAP6-dependent fashion and that AKAP6 is required for osteoclast resorption function; that study was performed in osteoclast derived from peripheral blood monocytes (10.7554/eLife.61669).

Here, the authors used the RAW264.7 mouse macrophage-like cell line, a widely used model to differentiate osteoclasts.

Here the authors claim that in RAW264.7-derived osteoclasts:

- MTOC and Golgi proteins at the nuclear envelopes (Figure 1)
- the nuclear envelopes, associated or not with Golgi, nucleate microtubules (Figure 2)
- express different isoforms of Nesprin1 and AKAP6 (Figure 3)
- that AKAP6 is required for recruitment of Golgi and MTOC components at the nuclear envelope, and for the nuclear envelope to organize microtubules (Figure 4)
- but that AKAP6 is not required for centrosomal microtubule nucleation (Figure 5)
- a reduced in the overlap between actin and microtubules at the actin ring in the absence of AKAP6 (Figure 6)
- reduced resorption activity in the absence of AKAP6 (Figure 7)

General comment:

Overall, the images are of good quality and the manuscript is clearly written.

Nevertheless, the present study recapitulates the former findings in human osteoclasts, without providing any new mechanistic insights into the link between microtubule nucleation at the nuclear envelope and the osteoclasts resorption function.

Several experimental approaches lack sufficient description, in particular the method of quantification by RT-PCR, the quantification of images using BIOP JACoP and the method to quantify the resorption activity, the quantifications in figures 6 and 7 lack robustness.

We thank the reviewer for the thoughtful and constructive evaluation of our manuscript. We appreciate the recognition of the image quality and clarity of the text and address the reviewer's concerns below.

We would like to note that while our previous publication (Vergarajauregui et al., eLife 2020) established that human monocyte-derived osteoclasts form an AKAP6-dependent NE- MTOC, the current study is not a replication of that work. Rather, the study addresses the question whether the NE-MTOC can be selectively disrupted without affecting the centrosomal MTOC, and further characterizes the NE-MTOC formation and function in osteoclasts providing several mechanistic advances. This is important, as it has recently been reported that centrosomal MTOC activity is required for the resorption activity of osteoclasts (see Introduction, line ???).

The main conceptual advances of the current manuscript include:

1. Identification of isoform-specific NE-MTOC components in osteoclasts
 - We show for the first time that RAW264.7-derived osteoclasts selectively induce nesprin-1 α , and AKAP6B, two isoforms previously considered muscle-restricted. These isoform-specific data were not examined in our prior human osteoclast study and represent a new regulatory layer in NE-MTOC formation during osteoclastogenesis. This finding provides biological insight into how osteoclasts activate the NE-MTOC program during differentiation.
2. Functional dissection of NE versus centrosomal MTOCs
 - The central goal of this manuscript was to determine whether AKAP6 depletion selectively impairs NE-MTOC activity or also affects centrosomal MTOCs. Using siRNA-mediated knockdown combined with washout assays and α -tubulin + EB1 imaging, we demonstrate that NE-derived microtubule nucleation is abolished, whereas centrosomal nucleation remains intact. This selective impairment is a key mechanistic advance and forms the conceptual foundation of the manuscript.

- Notably, we show that Griseofulvin, which was previously used as specific inhibitor of centrosome clustering, affects the NE-MTOC.
3. Quantitative analysis of microtubule-actin coordination at the sealing zone
 - We introduce a new quantitative imaging approach using co-localization analysis (BIOP JACoP) to examine the spatial relationship between microtubules and the podosome belt/sealing zone. This analysis provides novel insight into how loss of NE-MTOC function may lead to disorganization of the actin ring.
 4. Direct functional link between NE-MTOC disruption and impaired bone resorption
 - We combine calcium phosphate substrates, fluorescent calcein-based resorption assays, and simultaneous visualization of sealing zones, allowing the first direct single-cell-level correlation between NE-MTOC activity and sealing-zone-mediated resorptive capacity. This approach goes beyond the bulk resorption measurements used previously.

In response to the reviewer's concern regarding methodological detail and quantification robustness, we have implemented the following major improvements:

- All endpoint RT-PCR experiments have been replaced with quantitative real-time PCR (qPCR), now shown in Fig. 3B-D.
- All analyses involving image quantification (including co-localization analyses and resorption assays) now include:
 - o superplots showing biological and technical replicates,
 - o explicit descriptions of the quantification pipeline
 - o improved detail in the Methods section.
- The description of JACoP-based analysis has been expanded to include the thresholding strategy, replication structure, and statistical analysis.
- The resorption assay section now includes a detailed step-by-step description of the calcein-based quantification method, which allows simultaneous visualization of sealing zones and CaP degradation.

In summary, although our study builds on our prior identification of AKAP6 as an NE-MTOC component, it provides several new biological insights, including:

- isoform-specific induction of nesprin-1 α and AKAP6 β during osteoclastogenesis,
- direct functional separation of NE versus centrosomal MTOC activity,
- new quantitative analysis of microtubule-actin interactions at the sealing zone,
- direct functional coupling of NE-MTOC disruption to impaired resorptive activity at the single-cell level.

Collectively, we believe that these advances significantly extend the understanding of NE-MTOC biology in osteoclasts and clearly go beyond the findings of our previous work.

Major comments

- line 205-206: it is does not seem appropriate to conclude "These results suggest that pharmacological disruption of the clustering of centrosomes affects both centrosomal and nuclear envelope-associated MTOCs.", because Griseofulvin binds to tubulin and interferes with microtubule dynamics; thus, it is not possible to link Griseofulvin effects on microtubule nucleation to a defect in centrosome clustering.

We agree with the reviewer that the effects of Griseofulvin cannot be directly attributed to defects in centrosome clustering, as this compound primarily targets tubulin and disrupts microtubule dynamics. However, Griseofulvin has been previously used to demonstrate that centrosome declustering impairs osteoclast function (Philip et al., 2022). Our results indicate that its effects are not specific to centrosome clustering and also impact NE-associated microtubule organization. Highlighting this lack of specificity was precisely one of the points we intended to clarify in this study. We have now changed the text as follows:

"GF has been reported to inhibit centrosome clustering in osteoclasts resulting in impaired F-actin ring formation and bone resorption (Philip et al., 2022). However, while GF is reported to interfere with centrosome clustering (Philip et al., 2022; Rebacz et al., 2007), it is known that GF affects the dynamic instability of MTs (Panda et al., 2005). Consistent with this, GF treatment of RAW264.7-derived osteoclasts led to a noticeable redistribution of MTs, resulting in shorter, less organized filaments attached to both centrosomal and NE regions (Fig. S2). Furthermore, PCNT

staining at the NE shifted from a relatively homogeneous distribution to a more punctate, discontinuous pattern, indicating altered NE-MTOC architecture. These results suggest that pharmacological agents used to disrupt centrosome clustering may affect both centrosomal and NE-MTOCs. Therefore, alternative approaches will be necessary to distinguish the specific functional contributions of the two different MTOCs.”

- Figure 4A/lines 266-267: how was PCR performed to be able to quantify Akap6 mRNA? No protocol is specified in the methods section, and the results shown are end point PCR, not Q-PCR.

As also requested by Reviewer 1, we have replaced all end-point RT-PCR experiments with quantitative real-time PCR (qPCR) to provide a more rigorous and reliable quantification of AKAP6 and nesprin-1 isoform expression during osteoclast differentiation, as well as the efficiency of AKAP6 depletion. The updated qPCR data are now included in revised Figure 3B-D and Figure 4A. A detailed description of the qPCR protocol, including normalization strategy ($\Delta\Delta Ct$), and analysis pipeline, has been added to the updated Methods section (Lines 528-533).

- Figure 4A: comment on the fact that Syne1 giant mRNA is not detected here in osteoclast, as compared to the kinetics shown in 3B.

We thank the reviewer for this observation. In the original version of the manuscript, all PCRs were end-point RT-PCRs which are not quantitative, and the differences between the data in Fig. 3 and 4 may have reflected differences in the amount of cDNA loaded. To address this point, we have replaced all RT-PCR data with qPCR, and the previous end-point PCR panel from Figure 4A has been removed.

- Figure 5B: how was significance of the intensity differences assessed?

To assess the significance of the intensity differences shown in Figure 5B, we performed the following analyses:

- Peak position: For each nucleus, we determined the position of the highest α -tubulin intensity relative to the nuclear edge. Experiment-level means were then compared across biological replicates using a two-tailed t-test.
- Peak intensity: For each nucleus, we extracted the maximum α -tubulin intensity value, calculated experiment-level means, and assessed significance using a two-tailed t-test.

These details have now been added to the Methods section for clarity (Lines 612-614).

These quantitative results have been included in the revised text as follows: “In AKAP6- depleted cells, the α -tubulin intensity showed a significantly reduced intensity peak (1.60 ± 0.16 vs. 2.00 ± 0.13 in control, $P= 0.0264$) at the nuclear edge. Moreover, the peak position shifted further outward from the NE ($0.275 \pm 0.062 \mu\text{m}$ vs. $0.083 \pm 0.033 \mu\text{m}$ in control, $P= 0.0092$), indicating reduced α -tubulin growth from the nuclear envelope in comparison to control cells.” (Lines 314-317). The P values have been added to the graphs.

- Figure 5D: The increase appears around 30%, which is not "slight" (line 308).

We agree that, in general, a 30% difference would not typically be described as “slight.” However, centrosomal MTOC quantification in osteoclasts presents several technical challenges that limit the interpretability of this apparent increase. In particular, centrosomal aster length is difficult to measure reliably in osteoclasts because these cells exhibit heterogeneous centriole configurations (ranging from single to multiple centrosomes), as well as substantial spatial variation in centrosome positioning (e.g., located between nuclei or freely within the cytoplasm). As a result, direct quantitative comparison across cells is not trivial.

Additionally, small variations in nocodazole washout timing—which are experimentally difficult to control with absolute precision—can substantially influence the extent of microtubule regrowth and contribute to variability. When analyzing the data using biological replicate means (superplot approach), the increase is not statistically significant. Therefore, although there is a trend toward higher centrosomal MTOC activity upon AKAP6 depletion, we prefer not to overinterpret this observation. Therefore, we have modified our conclusion of this Results section to: “both together indicate that AKAP6 depletion does not inhibit centrosomal MTOC activity” (Lines 334-336).

- Figure 6: what is the BIOP JACoP macro, how was it used? How were ROI chosen? How many experiments were performed?

Thank you for indicating the lack of information and the misleading use of the term ROI. Because RAW264.7-derived osteoclasts are extremely large, they cannot be imaged at sufficient resolution in their entirety to analyze podosome structure. Therefore, we acquired images of random segments of the podosome belt and quantified the full belt region contained within each image.

To clarify this in the manuscript, we have expanded the Methods section to include the following description: “For quantifying actin-microtubule crosstalk at the podosome belt, high-resolution images of podosome belt segments were acquired. The belt region contained within each image was manually delineated using the polygon tool in ImageJ, based on the continuous F-actin ring. Colocalization analysis was performed using the BIOP JACoP plugin (Bolte and Cordelieres, 2006), a toolbox that computes pixel-based colocalization metrics (including Manders’ coefficients, Pearson’s correlation coefficient, and overlap fractions) between two fluorescence channels. For this study, we used the pixel overlap measurement (Manders’ coefficient M1) to quantify the fraction of the F-actin signal that colocalized with microtubules. Colocalization (overlap) values were expressed as arbitrary units (a.u.) representing the proportion of F-actin pixels overlapping with microtubule pixels relative to the total F-actin signal.” (Lines 615-625)

Figure 6B suggests it is only 1 experiment counting around 30 cells / ROI, in any case either it contradicts what is stated in the methods section lines 560-561 “All experiments were repeated at least three times.”, otherwise the p value would not be <0.0001. Of note, t-test is not appropriate for few experimental repeats, non-parametric test should be performed. The experimental approach must be clarified.

The reviewer is correct that the original graph suggested a single experiment. To address this, we now present the data as a superplot, where:

- each biological replicate (n = 3) is shown as a larger symbol in a distinct color,
- all individual cells (25-35 per experiment) are displayed as small points and colour-coded as per biological replicate,

This superplot format visualizes the number of experiments, their internal variability, and their reproducibility. The revised version now fully reflects the statement in the Methods that all experiments were performed at least three times.

Following recommended practice for superplots by J Cell Sci (Lord et al., JCB 2020, PMID: 32346721), we have analyzed the per-experiment means with an unpaired two-tailed t-test, which is valid for comparing group means across independent biological replicates. The effect size was large and consistent across the three experiments, resulting in a significant p-value. We now clarify this statistical approach in the Methods section (Lines 631-634).

What does “displayed less-defined F-actin rings” mean line 341, how was it assessed? Figure 6A does not suggest a less defined F-actin ring in the siAKAP6, neither does Figure 7A.

We thank the reviewer for the opportunity to clarify this point. F-actin staining was used to visualize the F-actin ring, the structural component of the podosome belt. In control osteoclasts, dense circular F-actin-positive puncta corresponding to podosomes were sharply defined and tightly packed. Our statement “displayed less-defined F-actin rings” meant that AKAP6 depletion resulted in F-actin stained rings in which the podosomes appeared less intense, less sharply resolved, and were surrounded by a more diffuse F-actin background. This observation was consistent and clear and thus we have decided for showing representative images as qualitative data (Lines 367-374).

We also noted that a similar “diffuse podosome organization” phenotype was reported in Maurin et al. (2021) following Tubb6 depletion, supporting the idea that disruptions in microtubule organization can impair podosome cohesion and maturation without abolishing the F-actin ring. While we had added this reference in the previous manuscript version, we have now clarified this point in more detail (Lines 371-374).

- Figure 7: same comments as above about experiment repetition and statistical analyses: the stats in 7B graph suggests that one cells was counted 1 experimental repetition. This has to be explained.

We thank the reviewer for these detailed comments. We have substantially revised the experimental description, quantification workflow, and statistical analysis for Figure 7 to address all concerns.

We now clarify in both the Results and the Methods that all CaP-resorption experiments were performed in three independent biological replicates. To make this explicit, the revised Figure 7C is presented as a superplot, in which individual sealing-zone measurements are shown and experiment-level means (color-coded per replicate) are used for statistical testing. Group comparisons were performed using a two-way ANOVA with Bonferroni correction, applied to the experiment-level means.

The quantification approach is odd. The methods section states lines 528-530: "The ratio of calcein intensity inside the sealing zone to that outside was calculated. A ratio below 1 indicated CaP resorption." How can the authors explain that the calcein ratio over 1 in cells with the smallest sealing zones and in all siAKAP6 sample? The mean/median should have a maximum around 1, not around 2; otherwise it suggests that osteoclasts accumulate calcium phosphate within the sealing zones. How/where was outside calcein intensity measured?

We agree that the initial description of the calcein-ratio calculation requires clarification. We have clarified this point in the Methods and refined the quantification to eliminate artifacts that previously inflated the signal inside the sealing zone.

Specifically, we made the following changes:

- The sealing zone was delineated using the phalloidin-positive F-actin ring, providing a precise definition of the "inside" region.
- Calcein intensity was measured strictly within the sealing-zone interior, avoiding border pixels where the signal is artificially elevated.
- The "outside" intensity was measured in an area of equivalent size immediately surrounding the sealing zone, while avoiding adjacent pits or background variations.
- These corrections eliminated the previously inflated ratios ($>1.5-2$).
- Despite all adjustments, we still observe increased calcein signal intensities in regions where resorption does not occur. The reason for this remains unclear. It is possible that, although CaP is not resorbed, it undergoes structural modification or compaction. Notably, calcein fluorescence is influenced by pH and by the organization of calcium phosphate, as it preferentially binds to and fluoresces in areas where CaP minerals accumulate.

The authors refer to Cen et al., 2022 for quantification, but the quantification approach is not as in that paper, which uses the classical method to measure the surface resorbed in the whole well, not below selected cells.

We have clarified the distinction between our approach and that of Cen et al. (2022) in the Results (Lines 381-387). Cen et al. established the method of fluorescent CaP labeling with calcein to visualize resorption pits, however, their quantification was based on whole-well resorption measurements. In contrast, our adaptation utilizes calcein to fluorescently label CaP to enable simultaneous visualization of sealing zones (phalloidin staining) and resorption pits by fluorescence microscopy, allowing us to quantify resorptive activity at the level of individual sealing zones. This approach makes it possible to directly correlate sealing-zone morphology with functional output within the same osteoclast.

To complement this sealing-zone-resolved analysis, we have now included von Kossa- based whole-well resorption data as the first panel in Figure 7 (Fig. 7A), allowing readers to compare the classical assay with our higher-resolution, cell-level quantification (Lines 380-381).

- About image quantification, lines 559-560: "The fluorescence intensity values for each band were normalized to the mean intensity of the whole cell." How was total fluorescence measured in osteoclast for normalization, provided that no entire cell is imaged in the present study.

We thank the reviewer for pointing this out. We took this description from our previous publication in which we indeed normalized to individual cardiomyocytes. Due to the cell morphology/size/multinucleation/nuclei aggregation of osteoclasts, this is here not possible. Consequently, the original sentence was incorrect, and we apologize for the confusion. No whole-cell normalization was performed. Instead, the fluorescence intensity values for each 0.2 μm band were normalized to the mean intensity of the entire -2 to $+2$ μm region surrounding the nuclear envelope, which corresponds to the region used for generating the intensity profiles. We have corrected this in the Methods section to accurately reflect the quantification procedure

(Line 612).

Second decision letter

MS ID#: jcs.264166R1

MS Title: Selective disruption of microtubule formation at the nuclear envelope impairs the bone resorption capacity of osteoclasts

Authors: Silvia Vergarajauregui; Samantha Panea; Jakob O Oltmanns; Ulrike Steffen; Felix B Engel
Article Type: Research Article

Dear Dr Vergarajauregui,

I am happy to tell you that your manuscript has been accepted for publication in Journal of Cell Science, pending standard publication integrity checks.